## Registered report

psychology

COVID-19, coronavirus, older adults, risk-taking, risk attitude

**Author for correspondence:**
Kelly Wolfe
e-mail: kwolfe@ed.ac.uk

# Age differences in COVID-19 risk-taking, and the relationship with risk attitude and numerical ability

Kelly Wolfe[1,2], Miroslav Sirota[1] and Alasdair D. F. Clarke[1]

[1]Department of Psychology, University of Essex, Wivenhoe Park, Colchester, Essex CO4 3SQ, UK
[2]Department of Psychology, School of Philosophy, Psychology and Language Sciences, The University of Edinburgh, 7 George Square, Edinburgh EH8 9AD, UK

 KW, 0000-0002-4077-6415; MS, 0000-0003-2117-9532;
ADFC, 0000-0002-7368-2351

This study aimed to investigate age differences in risk-taking concerning the coronavirus pandemic, while disentangling the contribution of risk attitude, objective risk and numeracy. We tested (i) whether older and younger adults differed in taking coronavirus-related health risks, (ii) whether there are age differences in coronavirus risk, risk attitude and numerical ability and (iii) whether these age differences in coronavirus risk, attitude and numerical ability are related to coronavirus risk-taking. The study was observational, with measures presented to all participants in random order. A sample of 469 participants reported their coronavirus-related risk-taking behaviour, objective risk, risk attitude towards health and safety risks, numerical ability and risk perception. Our findings show that age was significantly related to coronavirus risk-taking, with younger adults taking more risk, and that this was partially mediated by higher numeracy, but not objective risk or risk attitude. Exploratory analyses suggest that risk perception for self and others partially mediated age differences in coronavirus risk-taking. The findings of this study may better our understanding of why age groups differ in their adoption of protective behaviours during a pandemic and contribute to the debate whether age differences in risk-taking occur due to decline in abilities or changes in risk attitude.

## 1. Introduction

The new coronavirus (SARS-CoV-2; COVID-19) is a highly infectious disease that causes acute respiratory syndrome and has reached most countries around the world. On 30 January 2020, the World Health Organization (WHO) declared the outbreak a public health emergency of international concern. According to the WHO, the global number of deaths on 14

September 2021, stands at 4 636 153 deaths, of which 134 261 deaths have been recorded in the UK.[1] The UK government announced a nationwide lockdown on 23 March 2020, and a series of measures to prevent the spread of the virus. Staying indoors as much as possible, keeping others at a safe distance, exercising outdoors only once a day and washing hands often with antibacterial soap were measures the public was asked to adhere to in order to prevent further spreading of the virus and protect the National Health Service. Since then, restrictions and lockdown measures have been loosened or removed entirely, but the number of infections continues to soar.

Adherence to the government-mandated preventive measures is believed to be critical to curb the spread of the infection but there are individual differences in the extent people apply these preventive measures. Some UK citizens have openly protested the compulsory use of face masks in shops and public transport, with similar protests on mask usage in other countries such as Germany and the United States. A survey conducted on UK citizens during the first week of lockdown found that 60% of respondents reported following government guidelines completely, and 6% reported following guidelines only half the time or less [1]. Subsequently, a survey by University College London [2], which includes cross-sectional data from over 10 weeks, showed that guideline adherence in their sample had decreased by about a fifth, from 70% at the start of the survey to 50% at the end of May. However, this decline differed between age groups: while more than 6 out of 10 older adults reported following government guidelines entirely, only 4 out of 10 younger adults said to do the same. In studies on behaviour during prior epidemics, researchers found similar results; younger adults reported following guidelines less as well as perceiving less risk compared with their older counterparts, during the SARS epidemic in Canada in 2003 [3]. Additionally, a study on the 2009 influenza epidemic in The Netherlands found that older age was associated with higher intention to adopt protective measures [4]. Since not adhering to guidelines exposes the individual, as well as others, to risk, this behaviour can be considered a form of health-related risk-taking.

It is important to replicate and understand the nature of these age differences for theoretical and practical reasons. Firstly, from a theoretical perspective, investigating the contribution of risk perception, risk attitude and numerical ability to age differences in risk-taking adds to a growing body of work on older age and risky decision making. Older adults are generally considered to be more careful, especially when it comes to their health and safety. However, prior research on age differences and risk-taking has already shown that age-related risk-taking is highly dependent on context, such as framing, learning components and whether materials are description- or experience-based [5–7]. The current study adds to the existing research as it measures age differences concerning real-life risk during an unprecedented situation in our lifetime.

Secondly, there are practical reasons to investigate age differences in risk-taking.[2] At this time, COVID-19 has affected people's lives worldwide for over a year and a half. In that time, there have been multiple variants of the virus, some deadlier than others. Though vaccines are now available, the rate of infections remains high and though vaccinations offer protection against COVID-19, it is still possible to become ill as a result from a coronavirus infection, with some nations offering booster shots to those eligible. As such, it remains vital to understand what factors play a role in guideline adherence. These findings could benefit risk communication during the remainder of the pandemic as well as after, as it highlights what areas communication should focus on. For instance, if low numerical ability is associated with lower guideline adherence, risk communication could be improved by limiting the use of large, complicated numbers or figures. In addition, if younger adults report a lower likelihood of following government guidelines, communication about the virus can be tailored and sent through channels more specific to that age group to convey the risk of coronavirus more clearly.

We considered four factors, known to differ between older and younger adults, that could account for the observed differences in risk-taking between these age groups: objective risk for COVID-19 complications, risk perception, risk attitude and numerical ability.

Since the start of the outbreak in December 2019, there have been over 225 million coronavirus infections, and more than four and half million people worldwide have died.[3] To understand the workings of the virus, and identify who is most vulnerable, possible risk factors to COVID-19 are being

---

[1]The death toll at the time of the approved protocol (22 September 2020), stood at 1 155 235 deaths, of which 44 896 deaths were recorded in the UK.

[2]This paragraph has been altered after editorial approval on 13 September 2021. Originally, this paragraph discussed the possibility of a second wave of infections. As such, this paragraph was updated to reflect the relevance of this research in the current circumstances.

[3]At the time of the approved protocol (22 September 2020), the number of global infections was 40 million, and over a million deaths worldwide.

investigated. Studies on patients with coronavirus in China, where the virus was first reported, report a multitude of risk factors. A meta-analysis by Wang *et al.* [8] found that patients with comorbidities such as cardiovascular disease, hypertension, diabetes and chronic obstructive pulmonary disease (COPD) were more likely to experience severe illness as a result of coronavirus infection. These findings are further supported by Zheng *et al.* [9], who also found that respiratory illness was common among those with severe illness, and those who had died as a result of COVID-19. Studies on populations outside China found similar results, reporting that diabetes [10–14], cardiovascular disease [14,15] and COPD [10,11,14,15] were risk factors for severe coronavirus complications.

In addition to comorbidities, several personal characteristics have been found to increase the chance of coronavirus complications. For example, men have a higher chance of experiencing severe symptoms or dying as a result of coronavirus than women [9–11,14].

Ethnicity has also been found to impact the likelihood of complications [10,12,13]. Price-Haywood *et al.* [16] found that most patients who were hospitalized (76.9%) or died (70.6%) due to coronavirus complications were Black, despite only making up a little over a third of the study's Louisiana cohort.

However, older age appears to be one of the largest risk factors of coronavirus complications and mortality [9–16], with one study reporting people aged 80 or over having a more than 20-fold-increased risk compared with 50–59-year-olds [13]. Those most likely to die from coronavirus are those of older age, especially if they are male and have comorbidities [9].

In the months since the outbreak of the virus, it has been well-documented that the majority of younger adults experience mild symptoms, with only a small proportion needing hospitalization or having died as a result of coronavirus. However, older adults (aged 65 and older) make up the majority of hospitalizations and mortalities. This distinct difference in risk between age groups may (at least in part) explain differences in the adoption of preventive behaviours. It may be that younger adults are less inclined to adopt preventive behaviours as their chances of hospitalization or mortality are much lower than those of older adults.

In addition to objective risk, we also explored people's subjective perception of their risk. While objective risk is an indicator of how likely a negative outcome is to occur, people's perception of their risk can differ from their actual risk. An example of such dissonance was found by Katapodi *et al.* [17] in their meta-analysis, in which younger women reported higher risk perception of breast cancer than older women, despite older age being an established risk factor for breast cancer. In the context of the current pandemic, risk perception may play a role in the adopting of preventative behaviours. Someone could view their risk of coronavirus as high, which then increases their likelihood to adhere to guidelines and minimize their chances of contracting the virus, despite their low objective risk. A recent study by Bruine de Bruin & Bennett [18] found that those who perceived higher risks concerning coronavirus were more likely to adopt protective behaviours. These findings are similar to those of prior pandemics; van der Weerd *et al.* [4] reported that only risk perception was associated with the intent to adopt protective measures during the influenza A (H1N1) pandemic in The Netherlands.

Risk perception may also explain differences in health behaviours between age groups. Prior research shows that older adults perceived more risk and were more cautious than younger adults concerning health-related activities as well as ethical activities [19], and that self-reported risk perception in social, financial and recreational domains increased with age [20]. A study on differences in COVID-19 risk perceptions by Bruine de Bruin [21] found that older adults reported perceiving more risk of mortality if infected with COVID-19 but reported seeing less risk in getting infected or quarantined. These findings demonstrate the effect of people's subjective perception of risk on risk-taking behaviours, regardless of their objective risk. As such, this study will also examine people's perspective of their risk, in addition to objective risk, using exploratory analyses.

Second, individual attitudes towards risk can account for the age differences in risk-taking: people become more risk averse as they age. Risk attitude can be defined as the degree to which an individual appears to avoid or seek out risky options or behaviours [22]. Risk attitude goes beyond merely risk-taking, which is the likelihood of engaging in risky behaviour, as it incorporates other factors such as the person's perception of risk as well as the perceived benefit of the risky activity, and describes a more general disposition towards risk. Although one can have an overall risk attitude, indicating that an individual is generally more or less comfortable with risk, there is evidence that risk attitude also differs across domains such as health, social and recreational risk [23,24]. Though risk attitude is considered a stable psychological trait, it may change over time. Past studies have investigated the differences between younger and older adults in terms of risk perception, risk attitude and risk-taking behaviour by means of self-reports or through risk-taking in an experimental laboratory setting. There is evidence that people become more risk averse as they age [23,25], though

people's feelings towards risk may vary according to the domain. Rolison *et al.* [24] found that younger adults reported being more likely to take risks in the social domain, as well as health and safety, compared with older adults. Older adults were found to be more risk avoidant concerning health risks; they reported being less likely to undertake a health or safety risk, saw less benefit in these risks and reported higher risk perception than younger adults. These differences across domains are supported by Josef *et al.* [23], who reported declines in financial, driving, health, social and recreational risk-taking in older age, with differing rates of decline. As following guidelines is a key to preventing the spread of the virus, risk attitudes could provide more information on why people differ in how strict they adhere to guidelines. It may be that those who choose not to follow guidelines completely, whether in part or not at all, have higher risk-seeking attitudes concerning health. These individual differences in behaviour towards coronavirus may be (partly) explained by underlying, more stable personality traits concerning risk-taking.

Third, people's numerical ability may explain the age differences in risk-taking. At its core, numeracy encompasses the ability to do simple arithmetic operations and compare numerical quantities. However, higher numerical abilities also include logical and quantitative reasoning, and understanding concepts such as fractions, percentages, probabilities and proportions [26]. Those with lower numerical ability have been found to experience difficulties in judging risks, reading graphs, and are more sensitive to framing effects [26–28].

When examining the role of numeracy within the context of health-related risk, Petrova *et al.* [29] found that the effect of numeracy was a unique predictor to longer decision delays (i.e. time between symptom onset to decision to seek medical care), leading to significant increase in risk for death and serious disability. Participants with low numerical ability were four times more likely to delay critically needed medical treatment. Leiter *et al.* [30] found that those with low numeracy skills made worse patient prognostic estimates (participants were given case studies), as well as selecting treatments ill-fitting with patient prognosis (e.g. selecting an aggressive treatment for a 90-year-old man with 0% chance of survival or functional independence). Yamashita *et al.* [31] investigated numeracy and preventative health behaviours and found that low numerical ability was associated with lower likelihood of dental check-ups in older adults. Additionally, Peters *et al.* [32] found that lower numerical ability was associated with a lower willingness to take medication (participants were asked to calculate the likelihood of severe side effects prior to this, with the information provided to them).

At this time, daily counts of infections and deaths are given in newspapers and official briefings to inform the public how the virus is spreading and the progress of containment. However, simply providing numbers does not equate to understanding. A recent survey among UK citizens found that more than half of the working-age population has the numeracy level expected of a primary school child [33]. In the past months, news websites and TV programmes have been providing support in understanding what these numbers mean. In BBC's Coronavirus Special [34], numbers and graphs were explained to the public, as well as other news outlets publishing articles explaining what the coronavirus numbers mean and how to interpret them [35–37]. The ability to comprehend these numbers and apply them to calculate a useful statistic may influence people's willingness to take risks. Some may find these numbers confusing or difficult and may make miscalculations, which may cause misconception about the virus's severity, and may influence behaviour towards limiting the spread of the virus. However, this may vary between age groups. Current findings on age differences in numerical ability differ; some research has found no age differences in numeracy [28,38,39], while others have found that older age was associated with higher numerical abilities [33], or the opposite [28,40,41]. As numbers and graphs have been an integral part of risk communication during the pandemic, it may prove vital to understand how people's numerical ability influences their health behaviours during this time.

As older adults are considered one of the groups most at risk for coronavirus, while younger adults are generally considered to be most risk-taking, it is important to understand how these two age groups differ in their approach to the current pandemic. These differences, if present, may stem from a contrast in risk of coronavirus complications, their underlying attitude towards risk, or their ability to process and transform the numerical information given to them. So far, surveys and studies have been conducted to explore how people have behaved during the pandemic, and how much they have stuck to guidelines. However, no study has investigated what underlying, more stable factors such as risk attitude or numerical ability may explain age differences in health behaviours during the pandemic.

This study aimed to investigate how age differences in health-related risk-taking during the COVID-19 pandemic are related to objective risk, risk attitude and numerical ability. This has been addressed by the use of an online survey that included items on people's behaviour concerning

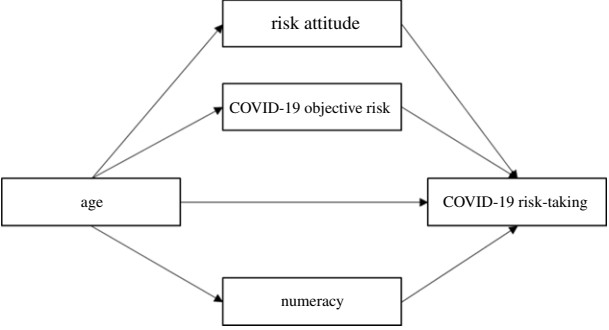

**Figure 1.** A visual representation of planned multiple mediation analysis.

guidelines, their (objective) risk of severe consequences of COVID-19 infection and questionnaires on risk attitude and numeracy. We hypothesized the following outcomes:

H1: age. Older adults will report following guidelines more often than younger adults, which is reflected in a higher mean in guideline adherence.

H1: objective risk. Those at higher risk of coronavirus complications will be more likely to adhere to COVID-19 guidelines and implement health measures.

H1: risk attitude. Those with an averse attitude towards health-related risk will be more likely to adhere to COVID-19 guidelines and implement health measures compared with those with a risk-seeking attitude.

H1: numeracy. Those with higher numerical ability will be more likely to adhere to COVID-19 guidelines and implement health measures compared with those with lower numerical ability.

If H1: age is not confirmed, we will not test any H2 below, and will continue with exploratory analyses instead. To test any H2, H1: age and any H1 matching the H2 must be confirmed. For example, to test whether the effect of age on COVID-19 risk-taking is mediated by objective risk, both H1: age and H1: objective risk have to be confirmed to continue with H2: objective risk, as those hypotheses concern the relationship between these two variables and COVID-19 risk-taking.

H2: objective risk. COVID-19 objective risk will mediate the relationship between age and COVID risk-taking. Older adults will be at higher risk than younger adults, which in turn will lead them to take less risk than younger adults.

H2: risk attitude. Risk attitude will mediate the relationship between age and COVID-19 risk-taking. Older adults will report a more risk-averse attitude towards health risks than younger adults and will take less risk relating to COVID-19 due to this.

H2: numeracy. Numeracy will mediate the relationship between age and COVID-19 risk-taking. Older adults having lower numerical ability than younger adults, which leads to them taking more risk relating to COVID-19 than their younger counterparts. For a visual overview of the planned mediation analyses, see figure 1.

# 2. Method

## 2.1. Power analysis

Prior to data collection, we conducted an *a priori* power analysis using a simulation-based approach. The direct effect of the age group on risk-taking was set to −0.3, with the effects of age group on risk attitude and numeracy also set to −0.3 and the effect of age group on objective risk set to 0.3. These three variables (risk attitude, numeracy and objective risk) were assumed to have an effect on risk-taking of 0.3 with the effects of objective risk and numeracy in the opposite direction to that of risk attitude (i.e. $y = 0.3 \times$ risk attitude $-0.3 \times$ objective risk $-0.3 \times$ numeracy). This allowed us to repeatedly simulate a dataset (500 times) for various sample sizes, for us to carry out the planned analysis. Based on these assumptions, and with $\alpha = 0.05$ and $1 - \beta = 0.95$, a sample size of $N = 400$ should suffice to verify all hypotheses. Please see electronic supplementary material, 1 for full code and details of the simulation and analysis, or visit https://osf.io/u6exb/ to access electronic supplementary material, 1 on the Open Science Framework.

We expected a 20% dropout rate (i.e. participants who have more than one measure incomplete. Therefore, we collected data from 480 participants (target $n = 400$, the expected dropout rate of 20% is equal to 80 participants) to obtain the analytical sample of $n = 400$.

**Table 1.** Variables included in the study.

| dependent variable | independent variable | descriptive variable |
|---|---|---|
| COVID-19 risk-taking | age group | trust in UK government |
| | COVID-19 objective risk | |
| | COVID-19 risk perception | COVID-19 numbers usage |
| | risk attitude | |
| | numeracy | |

## 2.2. Participants

A total of 489 participants took part in the study, 20 of which did not fulfil the inclusion criteria. The final analytical sample was 469 and consisted of 232 younger adults (49.6% identified as female, $M_{younger}$ = 26.52, s.d. = 5.16 years), and 237 older adults (49.8% identified as female, $M_{older}$ = 69.38, s.d. = 3.85 years). In the younger adult group, the majority listed a university undergraduate degree as their highest completed education (39.6%; followed by A-levels, 30%), that they were employed full time (49.6%; followed by the student, 25.4%) and had a household annual income of £30 001–£50 000 (33.6%; followed by £10 001–£30 000, 27.6%). Most younger adults reported that they had not been infected with COVID-19 (56.0%; followed by 'I'm not sure, but I don't think so', 26.7%). In the older adult group, the majority also listed university undergraduate as their highest completed education (29.5%; followed by both secondary school and A-levels, 25.7%), that they were currently retired (78.1%; followed by employed part-time, 10.1%) and reported a household annual income of £10 001–£30 000 (43.8%; followed by £30 001–£50 000, 31.6%). Most older adults also reported that they had not been infected with COVID-19 (79.7%; followed by 'I'm not sure, but I don't think so', 26.2%). Participant recruitment was done via Prolific Academic, with participants being paid £1.42 for taking part, with an hourly rate of £5.01 per hour, upon completion of the study. Only participants who (i) resided in England, (ii) fit the age criteria (aged between 18 and 35 years and aged 65 years or older) and (iii) had an approval rate of 90% were eligible to take part.

## 2.3. Materials and procedure

The materials included in the survey were given in a random order and were randomized within materials as well as between. All participants were given the same materials. The survey did not allow participants to skip items, and one item designed as an attention check was also included. The items included in the survey can be found in electronic supplementary material, 2, or at https://osf.io/u6exb/. All variables included in the study can be found in table 1, an overview of variables and materials used to measure them can be found in electronic supplementary material, table S1.

Participants were given a link to the survey via Prolific. In the study description, participants were told the general aim of the study and its prerequisites. Participants were told that they were not eligible to take part if they have been diagnosed with coronavirus. Those confirmed to have, or have had, coronavirus may approach the risks differently as it is widely assumed that antibodies will be present after recovery (for a period of time), and those cannot be infected again, or infect others. For this reason, people who have been confirmed to have (had) coronavirus were not included in the study. Participants were also not able to take part if they have been officially diagnosed with cognitive impairment, which was also communicated in the study description.

At the start of the survey, participants were given an information sheet with the details of the study, as well as a consent form. After providing consent, participants provided demographic information about themselves, including the county they reside in, education level, type of employment and annual household income. They also answered whether they believe they have, or have had, coronavirus, and if they have been officially diagnosed with cognitive impairment. These two items were included as screening items, in case participants did not read the study description on Prolific clearly. If participants answered yes to either of these, they were excluded from the analysis.

Following the demographic items, participants completed the objective risk stratification tool [42] to estimate their objective risk for COVID-19 complications. The measure is an existing risk assessment

measure, designed for the workplace assessment of healthcare workers. The items concern established risk factors for COVID-19, such as ethnicity, age, diabetes, pulmonary illness and cardiovascular disease. Answers to items may differ in the weight of their scoring, depending on the severity of the outlined illness. For instance, having diabetes type 1 or 2 without complications is scored as 1, while diabetes type 1 or 2 with complications (i.e. acute or chronic health problems, such as eye, foot and kidney problems) results in a score of 2, as diabetes complications increase the risk of severe disadvantageous outcomes of COVID-19 infection. Participants' total score is the sum of weights across all items, with higher scores indicating higher risk of severe complications resulting from COVID-19 infection.

Participants then completed 10 items concerning their behaviour in the current pandemic (e.g. 'Thoroughly cleaning my hands with hand sanitizer'). Six of the items reflect current government guidelines, such as wearing a mask on public transport and frequent handwashing, and four items concern recommendations such as using contact-free deliveries [43], not touching your face with unwashed hands and the use of hand sanitizer [44]. Though these recommendations are not part of official guidelines, the government has often communicated their importance to the public, as they help prevent infection of coronavirus. Participants were instructed, 'The next set of questions will present a number of activities and behaviours. You will be asked to report how often you have engaged in these behaviours in the last two weeks. Your answers will be fully anonymous, so please answer honestly.' They were then asked to rate how often they engaged in the outlined behaviours on a 5-point Likert scale ranging from 1 to 5 (1 = never, 2 = mostly not, 3 = sometimes, 4 = mostly yes, 5 = always). The option 'not applicable' is also included. Participants' risk-taking score is the arithmetic mean across all items, with scores near 5 indicating higher levels of risk-taking. As this is a novel measure, and has been designed for this study, we established its reliability using Cronbach's alpha. The scale's reliability was satisfactory, with an alpha of $\alpha = 0.73$. As such, risk-taking was measured as planned, through 10 items on preventative behaviours related to COVID-19.

Participants then expressed their perception of COVID-19 risk by completing the COVID-19 risk perception scale [45]. This six-item scale is measured as an index, covering affective, cognitive and temporal–spatial dimensions to provide a holistic measure of risk perception. The COVID-19 risk perception scale includes items concerning participants' perceived seriousness of the COVID-19 pandemic, perceived likelihood of contracting the virus themselves over the next six months, perceived likelihood of their family and friends catching the virus, and their present level of worry about the virus. Three of the six items are measured on a 5-point Likert scale (1 = strongly disagree, 5 strongly agree), the other three items are measured on a 7-point Likert scale (items 2–3: 1 = not at all likely, 7 = very likely and item 1: 1 = not at all worried, 7 = very worried). The pooled Cronbach's alpha across countries was $\alpha = 0.72$, the alpha for the UK sample was $\alpha = 0.80$. We equally found that the scale's reliability was good, with a Cronbach's alpha of $\alpha = 0.80$. Participants' risk perception was calculated by transforming the arithmetic mean for the six items to a value on a scale from 0 to 1, where higher scores nearest to 1 indicate higher risk perception.

Participants' risk attitude was measured by the 30-item domain-specific risk-taking scale (DOSPERT) [46]. This version is shorter than the original DOSPERT, and applicable to a broader range of ages, cultures and educational levels. Participants responded to six items concerning health and safety (e.g. 'driving a car without a seatbelt'), with identical items for each of the three subscales of the questionnaire (i.e. likelihood, expected benefits and risk perceptions). In the likelihood scale, participants rated the likelihood that they would engage in the given behaviours on a 7-point Likert scale from 1 to 7 (1 = extremely unlikely, 7 = extremely likely). In the benefit scale, participants rated the benefits that they perceived in the outlined behaviours on a 7-point Likert scale from 0 to 6 (0 = no benefits at all, 7 = great benefits). On the third scale, risk perception, participants rated the risk they perceived in undertaking the outlined behaviours on a 7-point Likert scale from 1 to 7 (0 = not at all risky, 6 = extremely risky). The internal consistency estimates (i.e. Cronbach's alphas) associated with the 30-item DOSPERT risk-taking scale ranged from $\alpha = 0.71$ to $\alpha = 0.86$, and those associated with the risk perception scale, from $\alpha = 0.74$ to $\alpha = 0.83$ [46]. We found that the reliability of the risk attitude scale was a Cronbach's alpha of $\alpha = 0.72$ (across all subscales), indicating sufficient reliability. Participants' scores on the risk attitude questionnaire were calculated by means of regressing the subscales risk benefit and risk perception on likelihood for each participant, using corresponding scores from each item which provided a (positive or negative) coefficient for each participant, in line with the recommended approach on the DOSPERT scoring sheet.

Participants' numerical ability was measured by the objective numeracy scale [47]. Participants were given 11 items for which they were required to calculate the answer to a mathematical problem (e.g. 'Imagine that we rolled a fair, six-sided die 1000 times. Out of 1000 rolls, how many times do you think the die would come up even (2, 4 or 6)?'). Participants were instructed 'You will be shown

**Table 2.** Three deviations from our preregistered approach.

| measure | type of deviation |
| --- | --- |
| attention check | We preregistered two attention checks in the study. Unfortunately, one of the attention checks was accidentally removed (as it was included with a replaced measure) in the last revision round. |
| COVID-19 risk-taking | We originally included the item 'meeting in groups larger than six people' but changed this to 'meeting indoors with people who are not in your household or bubble', due to the second lockdown in November 2020. This change was approved by the editor on 10 November 2020. |
| objective risk | The objective measure of risk required to ask about participants' sex at birth. However, our item measured their gender. To mitigate this confusion, we cross-checked our measured variable with our set requirements for participation in the Prolific Academic database (which included sex at birth). |

11 numerical questions. Each question will require you to calculate your answer. Each question has a few words in front of the answer line to indicate what type of answer is required. You may not use a calculator or any other means of help, except paper and pen for calculations (if needed).' The reliability of the objective numeracy scale, including the additional three items by Schwartz et al. [48], was found by Lipkus et al. [47] to be $\alpha = 0.78$. Weller et al. [28] reported a Cronbach's alpha of $\alpha = 0.76$, and Thomson & Oppenheimer [49] found a Cronbach's alpha of $\alpha = 0.72$. We planned for participants' score to be the sum of the number of correct answers. However, the scale's reliability was not sufficient, with a Cronbach's alpha of $\alpha = 0.68$. As this was below our stated cut-off of $\alpha = 0.70$, we first removed items in iterative ways, which did not improve the reliability of the scale. We then used an item-total correlation test to establish which of the 11 items was the best-suited item of the scale. The third item of the scale correlated most highly with the total score and has been used as an indicator of numeracy for the planned analysis instead of the full scale. However, we also conducted those analyses with the full scale (as well as the single item, as specified) in the Results section, so outcomes could be compared and for transparency. This is clearly mentioned in sections in which numeracy is included as a variable.

Lastly, participants were given two items on trust in the UK government's policies on coronavirus and how often they checked numbers on coronavirus deaths. These items are descriptive variables and were not included in the planned analyses.

## 2.4. Deviations from preregistration

Our accepted Stage 1 manuscript, unchanged from the point of in-principle acceptance, may be viewed at doi:10.17605/OSF.IO/3NV56. Here, we report three minor deviations from our preregistered approach (table 2).

## 2.5. Data processing

Participant data were eliminated if they answered 'yes' to one or more screening questions at the beginning of the survey (i.e. if participants have been diagnosed with coronavirus or cognitive impairment). Additionally, participants who answered the attention check incorrectly were also excluded. Missing data were treated as follows: if a small number of items (i.e. maximum of two items) within a measure had not been completed, the participant's score was calculated over the remaining items (i.e. instead of an average over eight items, it would be an average over six items). If more than two items of a specific measure had not been completed the measure was not included. If more than one measure was incomplete, the participant's data were removed entirely. The reliability of the scales used in the study was measured using Cronbach's alpha. If the reliability was unsatisfactory (an alpha below 0.7), we removed the items in iterative ways until we reached satisfactory reliability, or we used a single item instead (the item that is shown to be the best indicator of this measure).

## 2.6. Analysis

Data processing and statistical analyses were conducted in R (v. 4.1.0). To assess the reliability of the self-report measures, Cronbach's alphas were computed using the *ltm* package (v. 1.1–1), and item-total

**Table 3.** Descriptive statistics of risk-taking, objective risk and risk attitude in younger and older adults.

| measures | mean and standard deviation | | |
|---|---|---|---|
| | overall | younger adults | older adults |
| COVID-19 risk-taking | 1.92 (0.49) | 2.01 (0.51) | 1.84 (0.46) |
| objective risk | 2.54 (2.05) | 1.04 (0.94) | 4.00 (1.76) |
| risk attitude | −0.47 (0.71) | −0.50 (0.72) | −0.43 (0.70) |

correlations were computed using the *multilevel* package (v. 2.6). To choose the best predictors of risk attitude and cognitive ability, zero-order correlations were run using the *ggcorrplot* package (v. 0.1.3). For the mediation analyses, we used simple linear regressions to establish relationships between variables and then used the *mediate* function from the *Psych* package (v. 2.0.12) to run the mediation analyses. Each mediation model was run using a bootstrapping approach, with indirect effects computed for 500 bootstrapped samples. As such, the coefficients of the mediation analyses are accompanied by a 95% confidence interval.

## 2.7. Planned analysis

We planned to conduct regression analyses as part of a multiple mediation analysis. For the primary hypotheses (H1) we used four simple linear regressions with COVID-19 risk-taking as the dependent variable, and age group, COVID-19 objective risk, risk attitude and numeracy as predictors. If there was no effect of age group in the primary hypothesis (H1: age), we would stop the planned analysis and run exploratory analyses instead. If any of those primary hypotheses were confirmed, together with age (i.e. we established a direct relationship or a relationship between the mediator and dependent variable) we continued analyses to establish the required relationship between age and mediators (risk attitude, objective risk and numeracy), to then test the outlined H2 hypotheses in a multiple mediation model.

## 2.8. Exploratory analyses

The overall expectation was that younger and older adults differed in their risk-taking, and that this could be explained by age-related differences in objective risk, risk attitude and numerical ability. However, we were also interested in further exploring why people differ in their COVID-19 risk-taking. Using people's perception of risk as a mediator, we explored how the different types of risk perception (i.e. overall, for self and for others) mediated age differences in risk-taking.

# 3. Results

## 3.1. Confirmatory tests of hypotheses[4]

### 3.1.1. Age differences in COVID-19 risk-taking (hypothesis 1a)

We hypothesized that older adults would be less inclined to take risks than younger adults. Overall, we observed that our participants were not highly risk-taking, since the overall mean score was close to the lowest possible value of 1 (table 3). Nevertheless, younger people reported taking more risk than older adults, and as such had a slightly higher mean (table 3). Examining the differences at the individual item level, we can see differences for each item except mask wearing in shops and on public transport, which show flooring effects (figure 2).[5] To test our hypothesis, we used a linear regression, which revealed that the age group negatively predicted risk-taking, $B = -0.17$, $t_{467} = -3.77$, $p < 0.001$. As such, we confirmed our hypothesis that older adults took significantly less risk than younger adults.

---

[4]Our Stage 1 manuscript can be found at https://doi.org/10.17605/OSF.IO/3NV56.

[5]For an overview of the means and standard deviations for each item, separated by group, see electronic supplementary material, table S2.

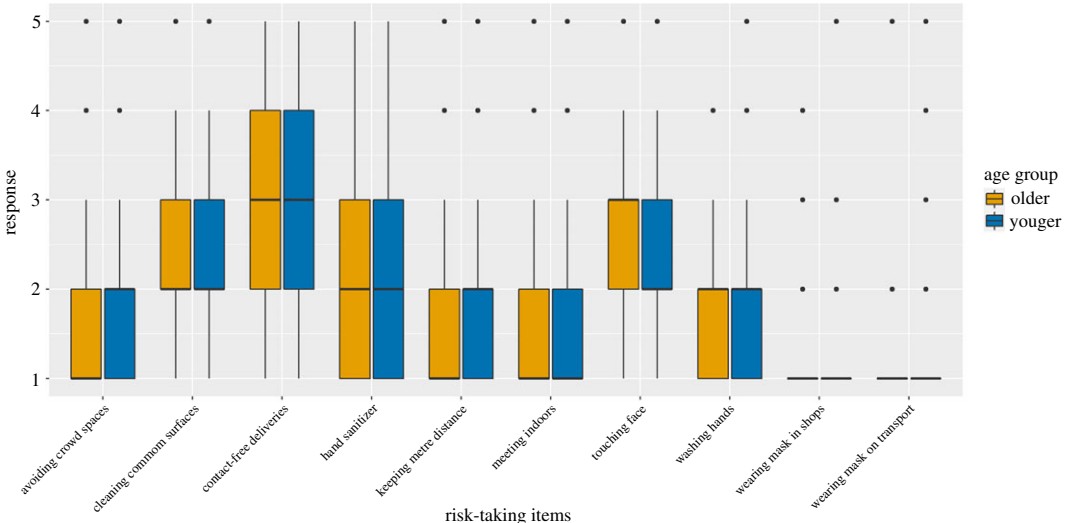

**Figure 2.** The distribution of all 10 items of the COVID-19 risk-taking scale, with separate distributions for older and younger adults. Note: significant differences were found between older and younger adults in their COVID-19 risk-taking. However, as group differences are small (i.e. one or two decimal points difference between means), the distributions of older and younger adults appear similar for most preventative behaviours in this figure. For an overview of the means and standard deviations for each item, separated by group, see electronic supplementary material, table S2.

**Table 4.** Zero-order correlations between COVID-19 risk-taking, age group, objective risk, numeracy (single item) and risk attitude.

| measure | 1 | 2 | 3 | 4 | 5 |
|---|---|---|---|---|---|
| 1. COVID-19 risk-taking | | | | | |
| 2. age group | $r = -0.17$ | | | | |
| | $p < 0.001$ | | | | |
| 3. objective risk | $r = -0.1$ | $r = 0.72$ | | | |
| | $p = 0.019$ | $p < 0.001$ | | | |
| 4. risk attitude | $r = -0.01$ | $r = 0.05$ | $r = 0.05$ | | |
| | $p = 0.864$ | $p = 0.284$ | $p = 0.331$ | | |
| 5. numeracy | $r = 0.21$ | $r = -0.13$ | $r = -0.05$ | $r = -0.05$ | |
| | $p < 0.001$ | $p = 0.007$ | $p = 0.382$ | $p = 0.268$ | |

### 3.1.2. Objective risk and risk-taking (hypothesis 1b)

We hypothesized that participants with higher objective risk of experiencing severe illness as a result of COVID-19 infection would be less likely to take coronavirus-related risk compared with those with lower objective risk. As expected, the mean objective risk was lower for younger adults than older adults (table 3). The objective risk was correlated most highly with age group, other correlations were of negligible effect size (table 4). To test our hypothesis, we used linear regression, which revealed that objective risk negatively predicted risk-taking, $B = -0.03$, $t_{467} = -2.36$, $p = 0.019$. As such, we confirmed our hypothesis that those with higher objective risk reported to take less coronavirus-related risk than those with lower objective risk.

### 3.1.3. Risk attitude and risk-taking (hypothesis 1c)

We hypothesized that those reporting a risk-seeking preference towards health and safety risk would be more likely to take coronavirus-related risk compared with those with a risk-averse preference. Overall, we observed that both younger and older adults reported being relatively risk averse, as both group

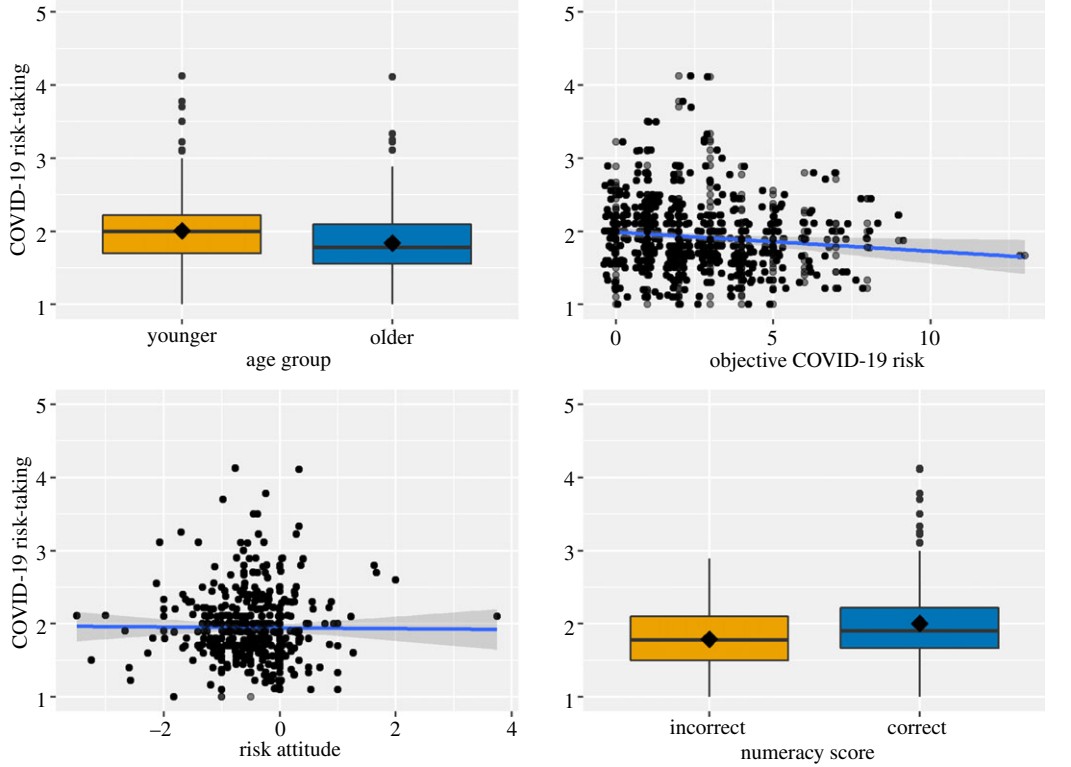

**Figure 3.** The relationships between COVID-19 risk-taking and age group, objective risk, risk attitude and numeracy. Note: both the distribution of age group and numeracy are displayed as box plots as these are dichotomous variables. Each boxplot has a line indicating the median risk-taking score, and a diamond shape representing the mean risk-taking score for that age group, or numeracy response. Objective risk and risk attitude are displayed as scatterplots, with risk-taking on the *y*-axis and objective risk or risk attitude scores on the *x*-axis. Each scatterplot also includes a regression line indicating the direction of the relationship between risk-taking and objective risk and risk attitude.

means were negative (table 3). Correlations of risk attitude with other variables were very small (table 4). To test our hypothesis, we used linear regression, which showed that risk attitude did not significantly predict risk-taking, $B = -0.01$, $t_{439} = -0.17$, $p = 0.865$. Thus, we did not confirm our hypothesis about the positive relationship between risk-seeking preference and risk-taking.

### 3.1.4. Numeracy and risk-taking (hypothesis 1d)

We hypothesized that those with lower numerical ability would be more likely to take coronavirus-related risk compared with those with higher numerical ability. Most participants answered the item correctly, with 62% of participants having given the correct answer. Younger adults were correct more often, with 69% compared with 57% of older adults. There was a small correlation between the numeracy item and risk-taking, other correlations were of minor size (table 4). To test our hypothesis, we used linear regression, which showed that numeracy positively predicted risk-taking, $B = 0.22$, $t_{467} = 4.69$, $p < 0.001$. Participants with higher numerical ability reported taking more risk (i.e. adopting fewer preventative measures) than those with the lower numerical ability (figure 3). The findings did not confirm our hypothesis, as we expected a significant relationship between numeracy and risk-taking but in the opposite direction. For better comparison with the results of other literature, we ran an identical analysis with the full 11-item scale. We found that the analysis yielded a similar outcome for this hypothesis, $B = 0.05$, $t_{467} = 4.26$, $p < 0.001$.

To summarize, we confirmed the hypotheses on the relationship between risk-taking and age group (1a), objective risk (1b) and numeracy (1d), but not on risk-taking and risk attitude (1c). In addition, the relationship between numeracy and risk-taking was in the opposite direction than we expected, with those having higher numerical ability taking more risk. As such, we tested the mediation hypotheses (H2 hypotheses), but only with confirmed mediators objective risk and numeracy (noted as 2a and 2c).

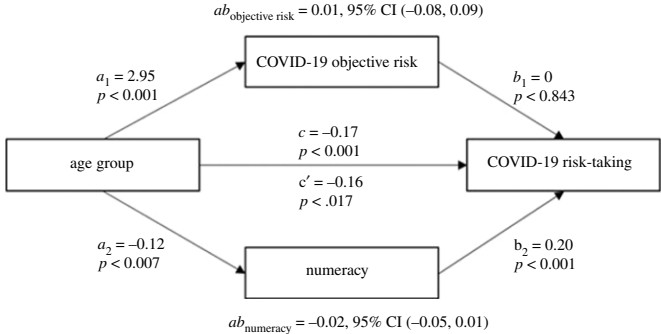

**Figure 4.** Mediation analysis on COVID-19 risk-taking, with age group as an independent variable, objective risk and numeracy as mediators. Note: the *a* path coefficient from independent variable to mediator, *b* path coefficient from mediator to dependent variable, *ab* path coefficient from independent variable to dependent variable via mediator (indirect effect), *c'* path coefficient from independent variable to dependent variable (direct effect), *c* path coefficient from independent variable to dependent variable (total effect). The reported confidence intervals represent 95% bootstrapped confidence intervals.

### 3.1.5. Age differences in objective risk, numeracy and their mediation of age differences in risk-taking (hypotheses 2a and 2c)

We hypothesized that older adults would be at higher risk of coronavirus complications, resulting in older adults taking less coronavirus-related risk than younger adults (2a). The mean objective risk was lower for younger adults than older adults (table 3). We first tested our hypothesis of a relationship between age group and objective risk (hypothesis 2a), using linear regression, which revealed that age group significantly predicted objective risk, $B = 2.95$, $t_{467} = 22.50$, $p < 0.001$.

We also hypothesized that numeracy would mediate the relationship between age and COVID-19 risk-taking, expecting lower numerical ability in older adults (2c). We used linear regression to test whether there were age differences in numeracy, which revealed that older age significantly negatively predicted numeracy, $B = -0.12$, $t_{467} = -2.70$, $p = 0.007$. We then proceeded by testing whether the relationship between age group and risk-taking was mediated by objective risk and numeracy through multiple linear regression, which included objective risk and numeracy as mediators. The results showed that objective risk no longer significantly predicted risk-taking, but numeracy did. We tested the significance of this indirect effect using a mediation model with objective risk and numeracy as mediators (figure 4). The results of the analysis indicated that the indirect effect of age group on coronavirus risk-taking through objective risk was not significant, thus disconfirming our hypothesis concerning objective risk (hypothesis 2a). When looking at numeracy, the mediation analysis demonstrated a significant indirect effect of age group on coronavirus risk-taking through numeracy, $ab = -0.02$, 95% CI [$-0.05$, $-0.01$], indicating partial mediation (figure 4). The findings confirmed our hypothesis concerning numeracy mediating the relationship between age group and risk-taking (hypothesis 2c). For better comparison with the literature, we also re-ran the same analyses with the full 11-item scale. We found no age difference in numeracy (when using the full scale), nor did numeracy mediate the relationship between age group and coronavirus risk-taking in the mediation model.

## 3.2. Exploratory analyses

In this section, we conducted a series of exploratory analyses that were not listed in the planned analysis. We were interested in how people's subjective perception of risk is linked with the adoption of preventative measures, and whether risk perception differed between age groups, numerical ability and types of risk perception (i.e. for self and others).

### 3.2.1. Mediation model with age group and risk-taking, with risk perception as a mediator

As objective risk did not mediate the relationship between age group and risk-taking in the planned analysis, we were interested in seeing whether the subjective perception of coronavirus risk would mediate age differences in risk-taking instead. Older adults reported undertaking less risky

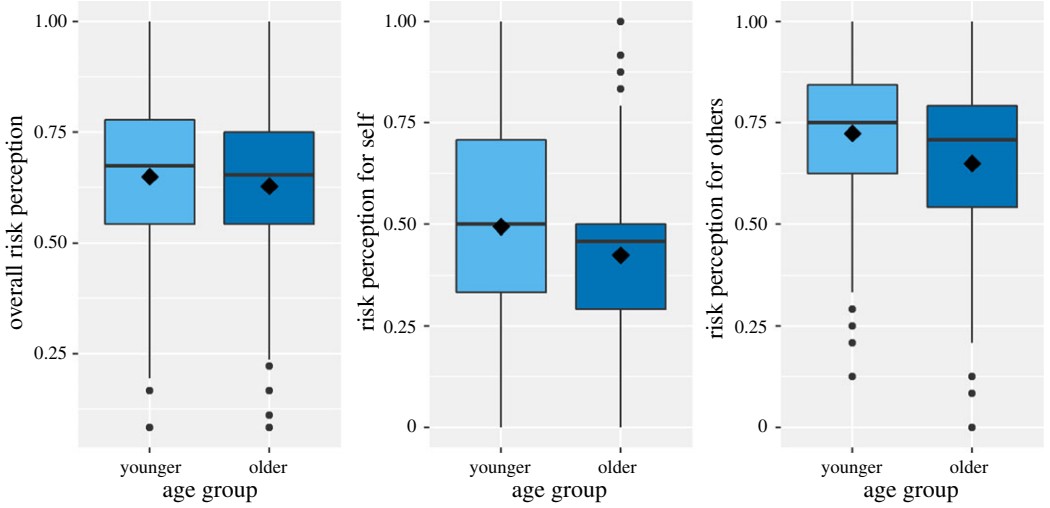

**Figure 5.** Age differences in risk perception overall decomposed in the perception of own risk and perception of others' risk. Note: from left to right: overall risk perception, risk for self and risk for others. The lines in the middle of the boxplots indicate the median risk perception, the diamond within the boxplot represents the mean risk perception for that specific age group.

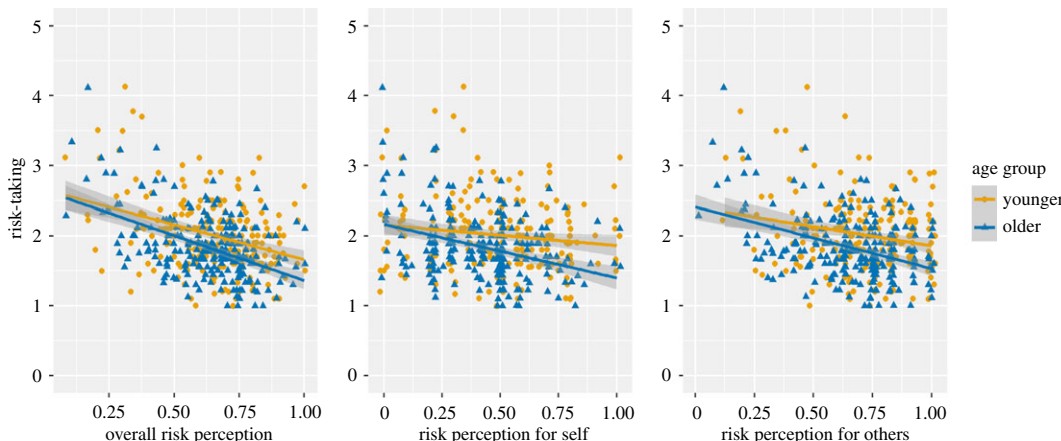

**Figure 6.** Scatterplots on risk-taking and its relationship with risk perception (overall, for self and for others) and age group.

behaviours than younger adults, which was characterized by a smaller mean in risk-taking (table 3). However, younger adults' mean risk perception was slightly higher than that of older adults (figure 5), suggesting that they perceived more risk than older adults. We tested whether age group and risk-taking were related using linear regression, with results revealing that age group negatively predicted risk-taking, $B = -0.17$, $t_{467}$, $p < 0.001$. Additional regression analyses suggested that risk perception negatively predicted risk-taking, $B = -1.10$, $t_{467} = -9.18$, $p < 0.001$, and that age groups did not differ in their perception of coronavirus risk, $B = -0.02$, $t_{467} = -1.35$, $p = 0.180$. Results of the mediation analysis, measuring the indirect effect of age group on risk-taking through risk perception, indicated that mediation was not significant, $ab = 0.02$, 95% CI [−0.01, 0.06]. These results suggest that risk perception does not mediate the relationship between age group and risk-taking behaviour.

The risk perception scale included questions that assessed COVID-19 perception more generally (e.g. 'Getting sick with the coronavirus/COVID-19 can be serious'), as well as questions that focused on personal risk of COVID-19 and the risk of others. Older and younger adults may not differ when accounting for all items, but they may differ in personal risk and others' risk. We, therefore, also calculated means for personal and others' risk (figure 5). Relationships between each type of risk perception and risk-taking differed across age groups (figure 6). The plots show visual differences between older and younger adults in terms of risk perception and risk-taking. As such, we decided to run two additional analyses using perception of personal risk and others' risk.

### 3.2.2. Mediation model with age group and risk-taking, with risk perception for self as a mediator

When only considering risk perception for oneself, we wondered whether older adults perceived their own risk of coronavirus as higher than younger adults, in line with their higher objective risk of coronavirus, and that their subjective perception of personal risk would mediate the relationship between age and risk-taking. Older and younger adults' reporting of risk perception did appear to be somewhat different, with younger adults having a slightly higher mean score (figures 5 and 6).

A linear regression on age group and risk-taking showed a significant difference in risk-taking across the two age groups, $B = -0.17$, $t_{467}$, $p < 0.001$, with older adults reporting less risk-taking. We then used a simple linear regression, regressing risk-taking onto the perception of personal risk. The results suggested that perception of risk for self significantly negatively predicted risk-taking, $B = -0.43$, $t_{467} = -4.33$, $p < 0.001$. We then regressed risk perception for self on age group, and found age differences in risk perception, $B = -0.07$, $t_{467} = -3.36$, $p < 0.001$, with older adults reporting a lower perception of personal risk. Following this, we ran a multiple mediation analysis with age group and risk perception for self as predictors of risk-taking. Both age group, $B = -0.20$, $t_{467} = -4.60$, $p < 0.001$, and personal risk perception, $B = -0.50$, $t_{467} = -5.08$, $p < 0.001$, remained significant predictors of risk-taking. We then checked for significant mediation effects using a bootstrapping approach, with age group, risk-taking and risk perception of self. Results suggested a significant indirect effect, $ab = 0.03$, 95% CI [0.01, 0.06], indicating that risk perception for self partially mediated the relationship between age group and coronavirus-related risk-taking.

### 3.2.3. Mediation model with age group and risk-taking, with risk perception of others as a mediator

We wondered whether participants who reported perceiving more risk for others would also report taking less risk, as wanting to protect others could be a reason for avoiding certain behaviours that increase coronavirus risk but did not have any prior expectations concerning age groups. Similar to risk perception of personal risk, younger adults' mean risk attitude score was higher than that of the older adult group, suggesting that younger adults perceived more coronavirus-related risk for others compared with older adults (figures 5 and 6).

We first regressed risk-taking on both age group and on perception of others' risk, using a simple linear regression, as part of the first step of the mediation analysis. Both age group, $B = -0.17$, $t_{467} = -3.77$, $p < 0.001$, and perception of others' risk, $B = -0.63$, $t_{467} = -5.88$, $p < 0.001$, significantly negatively predicted risk-taking. We then used a simple linear regression to regress perception of others' risk on age group. The results indicated that the age group negatively predicted perception of others' risk, $B = -0.07$, $t_{467} = -3.94$, $p < 0.001$. We then proceeded to the mediation analysis. First, we used a multiple linear regression with age group and perception of others' risk as predictors of risk-taking, and found that both age group, $B = -0.22$, $t_{467} = -5.12$, $p < 0.001$, and perception of others' risk, $B = -0.72$, $t_{467} = -6.86$, $p < 0.001$, were significant. We then tested for mediation effects using a multiple regression analysis with a bootstrapping approach, identical to prior mediation analyses. The model's indirect effect was significant, $ab = 0.05$, 95% CI [0.02, 0.09], suggesting that risk perception for others partly mediates the relationship between age group and coronavirus risk-taking.

### 3.2.4. Checking coronavirus statistics and dissatisfaction with UK coronavirus approach

Lastly, we examined how often people reported checking coronavirus statistics (i.e. numbers of infection, hospitalization and deaths relating to coronavirus) and dissatisfaction with UK COVID-19 policies. Checking coronavirus statistics was presented as a statement, with values closer to 7 indicating a higher level of disagreement (figure 7). Results of an independent sample $t$-test with age group and regularly checking coronavirus statistics indicated that older adults ($M_{older} = 2.98$) reported checking statistics more often than younger adults ($M_{younger} = 3.53$), $t_{458} = 3.3$, $p = 0.001$, $d = 0.31$. When looking at dissatisfaction with the UK's coronavirus approach, an independent sample $t$-test with age group and coronavirus approach showed age differences between groups, with younger adults reporting higher dissatisfaction with the UK government's coronavirus approach ($M_{younger} = 4.94$) than older adults ($M_{older} = 4.33$), $t_{458} = 4.4$, $p < 0.001$, $d = 0.41$.

Overall, the results suggest that age group, objective risk and numerical ability were all significant predictors of coronavirus-related risk-taking. When included together in a mediation model, the predictive power of objective risk disappeared, but numeracy partially mediated the relationship between age group and risk-taking. In the exploratory analyses, we found that though risk perception

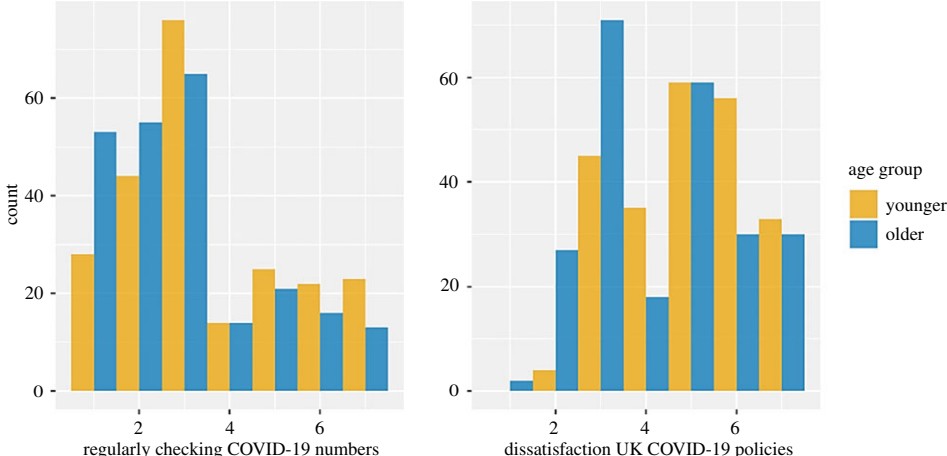

**Figure 7.** Dissatisfaction over UK COVID-19 policies (left) and regularly checking COVID numbers (right), separated by age group. Note: on the x-axis, participants' choice options are displayed, the y-axis displays the number of participants who chose the specific options. For dissatisfaction over UK COVID-19 policies (left), the options ranged from 1 (extremely satisfied) to 7 (extremely dissatisfied). For regularly checking COVID numbers (right), the options ranged from 1 (strongly agree) to 7 (strongly disagree).

was significantly related to risk-taking, overall risk perception did not differ between age groups. However, when only looking at items that reflected personal risk or risk for others, age groups did differ, with younger adults reported perceiving more risk for themselves as well as others compared with older adults.

## 4. Discussion

The current COVID-19 pandemic has been a major health risk, claiming 4 636 153 lives worldwide, 134 261 of those lives in the UK [50] at the time of writing this discussion (14 September 2021). While vaccination efforts in some countries, including the UK, are generally successful, the virus is still spreading globally, with the risk of new variants still present. As such, it is essential to understand the predictors of the adoption of protective behaviours to prevent further illness and mortality. We tested the assumption that younger adults take more risk than older adults, while testing three possible variables that could account for such a difference: differences in objective risk, risk attitude and numeracy. We confirmed that older people adopted more preventative measures, but, against our expectations, objective risk and risk attitude did not mediate this relationship. However, numeracy did partially mediate the relationship between age group and risk-taking behaviour, with younger adults scoring higher in numerical ability and reportedly taking more risk than older adults.

Our findings on age differences in the adoption of preventative measures are in line with other research on this topic, which showed that younger adults were less likely to implement preventative measures [18,51–56]. Why these age groups differ in the coronavirus-related risk they take could be due to the difference in risk of coronavirus complications. That younger adults are less at risk of coronavirus complications has often been communicated through the media [57] and government briefings. In the UK, the NHS only lists those aged 70 or older at moderate risk [58]. In addition, mortality rates provided by the UK government suggest that the proportion of coronavirus-related deaths of people aged between 15 and 44 years accounts for 1% of deaths, while the mortality of those aged 65 or older accounts for close to three quarters of total deaths. The large difference in risk in terms of hospitalization and death because of coronavirus infection may lead to younger adults taking more risk by not adopting preventative measures as often, as it is less likely that they will experience serious health-related consequences.

Another possible reason for this age difference may be differences in financial status and work environments. Financial concerns, such as the loss of income and job insecurity, are an often-reported concern for younger adults [53,55]. In the current study, half of those aged between 18 and 35 reported working full time. The UK government has asked citizens to work from home where possible but has equally allowed companies to decide whether employees are to work on location. Unlike the older age group, who largely reported being retired, it may be that the younger adult group are not able to consistently avoid crowded spaces, such as offices, public transport, schools or

supermarkets, and are unable to consistently socially distance with at least 1 m between themselves and others at all times. However, it is important to note that age differences in risk-taking were small, and both groups reported relatively low risk-taking.

Numerical abilities were a significant predictor of COVID-19 related risk-taking, with those having higher numerical abilities less likely to adopt preventative behaviours, thus taking more risk. This finding was the opposite of what we expected, as prior research has found that low numeracy was related to poorer health outcomes and decisions [29–32]. When looking at numeracy and COVID-19 health behaviours specifically, other studies found that numeracy was not significantly related to adopting preventative measures [56,59]. The conflicting findings may be explained by the presence of other factors involved in decision-making concerning COVID-19. For example, high levels of worrying are associated with higher adoption of preventative behaviours, people who are susceptible to misinformation are less likely to adopt preventative behaviours [56], and trust in science [45,60], political affiliation and risk perception [61] also influence the adoption of preventative behaviours. As such, understanding of the numerical information associated with coronavirus risk may play a lesser role compared with other factors in one's adoption of preventative measures.

We also assessed age differences in numeracy and found that older adults had lower numerical abilities compared with younger adults. This finding is not surprising, as there have been other studies that found lower numeracy among older adults [28,40,41,62]. Overall, our results indicated that younger adults took more coronavirus risk, which was partially mediated by their higher numerical ability, with higher numeracy associated with higher risk-taking behaviour. In a review on numeracy and risk literacy, Garcia-Retamero *et al.* [63] state that highly numerate individuals often have more accurate perceptions of health-related risks and benefits, and often make better decisions (leading to higher health benefits), compared with low numerate individuals. Unlike situations in prior work on health behaviours and numeracy, better understanding of the probabilities associated with COVID-19 could lead to more risk-taking among younger adults. For instance, the likelihood of hospitalization or death as a result from a coronavirus infection is much lower in younger adults than older age groups. As such, the higher numerical ability could translate to a better understanding of the probabilities associated with COVID-19, and as such, less adoption of preventative behaviours in younger adults, as it is less likely to lead to serious health consequences in younger adults.

The objective risk was found to be significantly related to COVID-19 risk-taking, which was in line with our expectations. Other studies, such as the COVID-19 Social Study, reported that those at higher objective risk did not adopt preventative measures more often than those at low risk of coronavirus-related complications [56,59,64]. Though we did find that those with higher objective risk adopted preventative measures more often, both those at low and high objective risk reported adopting preventative measures often (figure 3) and differences were small. This may be in part due to when this study was conducted. As data were collected during the second national lockdown, many of these behaviours were expected or specifically communicated by the UK government. As such, many people, both at high and low risk, may have adopted these behaviours as it was part of a national lockdown. Further research is needed to investigate why those at higher risk do not appear to adopt unenforced preventative measures more often than those at low risk, with data collected outside a national lockdown.

Risk attitude showed no relationship with adopting preventative measures in the study. The DOSPERT [46] is a measure of risk attitude in which people are asked whether they would engage in risky activities, as well as the benefit and risk they see in those activities. Items in the questionnaire are hypothetical situations, such as taking a ride in a taxi without a seatbelt. Though the negative consequences of those hypothetical situations can be severe, they are more everyday situations, in contrast to the current coronavirus pandemic. In addition, the coronavirus-related preventative measures in this study were communicated by the government and many of these measures are enforced, such as wearing a face mask on public transport. It may be that the risk attitude items are too distinct from the non-hypothetical risk posed by coronavirus to explain preventative behaviours during the pandemic. For future research, it could be beneficial to include a risk attitude measure that more closely resembles the decisions or behaviours people experience during a major health crisis such as COVID-19.

In the exploratory analysis, risk perception was a significant predictor of adopting preventative measures, like prior research [21,45]. Although older adults reported taking less risk (i.e. they adopted more preventative measures), there were no age differences in overall COVID-19 risk perception. We then also investigated risk perception for oneself and for others, as prior research has found that people's risk perception can depend on who the risk is perceived for (e.g. oneself, close friends,

family or strangers) [65,66]. When only examining items on personal risk and risk of others, younger adults reported perceiving more risk for both themselves and for others such as family and friends. As such, it is likely that other factors, beyond those included in this study, are involved in why younger adults adopted preventative measures less despite their high perception of risk. During the first national lockdown, instituted in March 2020, adults 70 years and older were advised to shield (i.e. not leaving one's house, and relying on other services to obtain supplies or essentials) [67], but this was no longer advised for the second lockdown, during which this study was conducted. As such, active shielding because of government advice cannot account for age differences in risk perception concerning coronavirus. Instead, these differences in perception may be due to age-related differences in circumstances, such as work environment or their well-being. According to the Office for National Statistics [68], younger adults reported feeling lonely more often than those aged 60 years and over, as well as reporting that COVID-19 had affected their work through reductions in hours worked and concerns about health and safety at work. A quarter of those young adults reported concerns on the impact of COVID-19 on their well-being. Other studies have similar findings, including older adults reporting less concern about their finances and mental well-being compared with younger adults [21,53,69]. Despite the high perception of risk for themselves and others, it may be that younger adults' perception of other stressors, such as their mental well-being or finances, drives these age differences in coronavirus-related risk-taking, despite younger adults' higher risk perception for themselves and others. Research by Schneider *et al.* [60] found that a combination of psychological factors, such as personal values, beliefs and the extent to which people believe they are able to contribute to curbing the spread of coronavirus shaped COVID-19 risk perceptions [60].

Future studies should explore which other variables underlie age differences in COVID-related risk-taking, and further examine why those at higher risk of COVID-19 consequences do not differ in the adoption of general recommendations to those at objectively lower risk of COVID-19 complications.

## 5. Limitations

Of course, this study is not without its limitations. Firstly, adoption of preventative behaviours was self-reported; real-life behaviour may differ from participants' self-reported behaviour due to social-desirability bias. Secondly, some of the items were designed with generalizability in mind, such as 'avoiding crowded spaces, as being too specific could make it difficult to answer or the item would be not applicable. This also had its limitations. For instance, it is not possible to determine whether participants indicated that they are not avoiding crowded spaces due to a lack of concern, or whether that is due to a lack of possibility to do so (e.g. public transport to work). Lastly, although we found significant relationships that explained differences in adopting preventative measures, it is likely that additional factors exist beyond those included in the study, such as financial, mental health [21,53,69] and trust in science [45,60], and we suggest that future research explores these further.

## 6. Conclusion

We aimed to further the understanding of whether age differences in COVID-19 risk-taking, characterized by the adoption of preventative measures, are related to numeracy, objective risk and COVID-19 risk perception. As COVID-19 is predicted to be present for the foreseeable future, we will have to rely on preventative measures such as social distancing to keep ourselves and others safe. Understanding what factors play a role in adopting preventative measures, and whether these mechanisms differ across age groups is crucial to prevent further illness and mortality.

Ethics. This study was approved by the Psychology Research Ethics Committee at the Department of Psychology at the University of Essex. Informed consent was obtained prior to participation in this study.

Data accessibility. Materials related to this manuscript, including data and scripts for analysis, have been uploaded to the Open Science Framework: https://osf.io/u6exb/.

The data are provided in the electronic supplementary material [70].

Authors' contributions. K.W., M.S. and A.D.F.C. conceptualized the study. K.W., M.S. and A.D.F.C. designed the instruments. K.W. collected and processed the data. K.W. conducted all of the analyses with input from A.D.F.C., and M.S. K.W. drafted the manuscript with input and revisions from M.S. and A.D.F.C. All authors gave final approval for publication.

Competing interests. We declare we have no competing interests.

Funding. This work was supported by the University of Essex Psychology Department Postgraduate Fund.

**Acknowledgements.** We thank Dr Alexandra Freeman and Dr John Kerr from the Winton Centre for Risk and Evidence Communication for providing us with numeracy scores that were used in the power simulation of this project.

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
