## [Peer Review File · Royal Society Open Science]

Review History

RSOS-201445.R0 (Original submission)

Review form: Reviewer 1

Do you have any ethical concerns with this paper?

No

Recommendation?

Accept with minor revision

Comments to the Author(s)

The aim of this study is to better understand why younger and older adults differ in their adherence to COVID-19 related public health guidelines. Specifically, it is proposed that risk attitude, risk perception and numerical ability will mediate the relationship between age and adherence to health guidelines ('COVID-19 risk taking'). While the questions outlined in this Stage 1 manuscript are timely, I have questions and comments about the rationale, method and analytical approach.

1. There needs to be further justification for the implied causal paths in the model. What, for example is the rationale for numeracy being directly related to risk taking? What is numeracy is only related to risk taking via risk perception?
2. The model assumes that the paths from age to the three mediators is equal in both age groups. How can this assumption be justified?
3. I was surprised to see age as a categorical variable. Why not model age as a continuous variable. What justifies the use of these two age groups and is the use of a categorical independent variable problematic for the mediation analysis?
4. I do not think that averaging the survey items that you believe reflect the mediators and the dependent variable is an optimal approach here. What if what you think are indicators of the same underlying construct turn out not to be? Then your mean scores will not be particularly meaningful. Relatedly, what if there are differences in the reliability of different survey items? To address such potential issues, it would be useful to take a latent variable approach and use SEM.
5. You do not provide any justification for the effect size you used to estimate statistical power. How did you arrive at this effect size? Also, there is no link to the cited supplementary information.
6. The choice of the numeracy measure is not clearly justified. In the introduction you talk about the importance of understanding proportions, percentages, fractions etc. It is not clear whether the 'Objective Numeracy Measure' measures aspects of numerical understanding and skills that are relevant to understanding COVID-19 data. In other words, numeracy may not uniformly relate to adherence to public health advice but instead it may be a subset of numeracy skills (in particular being able to understand, process and calculate with proportions) that are particularly relevant. Are you sure you are measuring aspects of numeracy that have previously been found to be associated with health-related behaviours?
7. The description of the survey items should be more detailed and a link to a document containing all the survey items should to be provided.

Review form: Reviewer 2

Do you have any ethical concerns with this paper?

No

Recommendation?

Major revision

Comments to the Author(s)

Wolfe and colleagues propose to investigate mediating factors affecting the relationship between age and Covid-19 related risk-taking behavior. The team has taken great care in presenting the background, hypotheses and especially methodology and data analysis plan. However, the first two elements, the background and hypotheses, need more consideration. The motivation behind the study is not entirely clear, given the hypotheses posed. While there is an urgent need to understand the pandemic's various ramifications, studies need to be grounded in objective reference frames to encourage public trust in research.

1. The scientific validity of the research question(s)

The most problematic issue in this registered report is the mishandling of the term risk. In particular, there is an assumed, implied and unverified relationship between Covid-19 preventative behaviors, risk “aversion” and rationality. However, a considerable debate is ongoing regarding the true, objective (country, age and health-condition adjusted) risk of infection; hospitalization; and mortality. What is the reference? Where are these objective measures in the report? Without that standard, measuring differences in Covid-19 risk-perception and risk-taking becomes a creative, not scientific, exercise.

Indeed, the problem of risk in relation to Covid-19 is that it remains difficult to compute. How do the authors plan to address this?

Further, a remarkable feature of Covid-19 related risk is in its non-uniform distribution, particularly with regards to age. How will the study disentangle true, individual-specific risk from individual perceived risk?

Here is an example of the flaw in the study’s premise:

P 4. line 48-49 “First, risk perception can account for the age differences in risk taking; older adults have a higher risk associated with contracting COVID-19 than younger people.” This statement encapsulates the main flaw in the study; risk perception and risk are two different things. They are confounded here. Both risk perception and true risk may differ between age groups.

All in all, I am not entirely convinced by the age factor in the model. More interesting would be the numeracy effect on risk-taking etc.

2. The logic, rationale, and plausibility of the proposed hypotheses

Without addressing the key premise of objective risk, it is difficult to judge the logic of the proposed hypotheses. However, the explicit value of finding a significant relationship between two related variables (eg risk perception and risk taking; and risk attitude and risk-taking) is not obvious.

p.8 line 52: Why is the relationship between risk perception and risk-taking hypothetical?

p.8. idem line 59 regarding risk attitude and risk-taking.

3. The soundness and feasibility of the methodology and analysis pipeline (including statistical power analysis where applicable)

The above are detailed and appear sound, however, the hypothesis in p.9. lines 7-12 may be tenuous. Here, you could cast a different model where numeracy predicts risk-taking and perhaps probe the mediating effects of age on that relationship. But this proposition still does not account for the disparate risk of Covid-19 on different age groups. Finally, why is the study constrained to the UK? Is this due to the Ethics approval? If it is not, it would be beneficial to open the study to other countries, particularly given the unique opportunity to study a global, rather than local, state of distress.

4. Whether the clarity and degree of methodological detail would be sufficient to replicate exactly the proposed experimental procedures and analysis pipeline

The methodology is well detailed and replicable. Only one minor concern regarding measures of Covid-19 related risk perception:

p.12, lines 20-24 It is not clear how the items are scaled, or what they are scaled by, before being averaged. Are estimated risks of infection, hospitalization and death combined into one measure per respondent?

5. Whether the authors provide a sufficiently clear and detailed description of the methods to prevent undisclosed flexibility in the experimental procedures or analysis pipeline

The authors methods description are transparent, including both the code and questionnaire, leaving little room for methodological subterfuge.

6. Whether the authors have considered sufficient outcome-neutral conditions (e.g. positive controls) for ensuring that the results obtained are able to test the stated hypotheses

This may remain open to interpretation and may also not be feasible given the comparative rarity of a global pandemic. Indeed, hypotheses 1-3 could be viewed as sanity checks, with the 4th hypothesis yielding the most pertinent results,

Minor comments:

p.1, line 60, this is not a mortality rate as it is not scaled to any standard.

p.5 lines 23-24: "Prior research shows that older adults perceive more risk in health-related activities than younger adults" Please provide a reference.

p.5 lines 26-33 Do older adults perceive less risk of infection than do younger people? Or do they believe they have less of an infection risk than younger people do? It is not clear here.

p.5 line 33-35 Please include some references here "Risk perception is a known predictor of risk behavior and has shown to vary between age groups".

p.6 lines 35-36 Please remove redundancy here.

p.8 line 24: In the second hypothesis, hasn't this already been reported in another study above?

Decision letter (RSOS-201445.R0)

Dear Ms Wolfe,

The Editors assigned to your Stage 1 Registered Report ("Age differences in COVID-19 risk-taking, and the relationship with risk attitude and numerical ability.") have now received comments from reviewers. We would like you to revise your paper in accordance with the referee and editors suggestions which can be found below (not including confidential reports to the Editor). Please note this decision does not guarantee eventual acceptance.

Please submit a copy of your revised paper within three weeks (i.e. by the 23-Sep-2020). If we do not hear from you within this time then it will be assumed that the paper has been withdrawn. In exceptional circumstances, extensions may be possible if agreed with the Editorial Office in

advance. We do not allow multiple rounds of revision so we urge you to make every effort to fully address all of the comments at this stage. Your manuscript is likely to be sent back to one or more of the original reviewers for assessment.

When submitting your revised manuscript, you must respond to the comments made by the referees and upload a file "Response to Referees" in "Section 2 - File Upload". Please use this to document how you have responded to the comments, and the adjustments you have made. In order to expedite the processing of the revised manuscript, please be as specific as possible in your response.

on behalf of Professor Chris Chambers (Registered Reports Editor, Royal Society Open Science)
openscience@royalsociety.org

Associate Editor Comments to Author (Professor Chris Chambers):

The manuscript has now been assessed by two expert reviewers. While both reviewers find areas of merit in the proposal, they also raise some significant concerns that will need to be addressed to achieve Stage 1 in-principle acceptance. The foremost issue to address is the rationale and conceptual basis of the study, which both reviewers question in different ways (Reviewer 1, point 1 and point 2; Reviewer 2 point 1). In addition, the reviewers note a range of methodological points where greater detail and justification are required for specific analytic choices and assumptions.

From an editorial perspective, this submission falls close to the line for an outright rejection due to the severity of concerns surrounding Stage 1 criterion 1 (scientific validity of the research question). Major concerns in this area often require substantial amendments to the primary objective of a study, which authors may not feel comfortable in making. That said, I think there is sufficient potential overall to offer the authors the opportunity to address these issues if they are willing and able to do so.

Comments to Author:

Reviewer: 1

Comments to the Author(s)

The aim of this study is to better understand why younger and older adults differ in their adherence to COVID-19 related public health guidelines. Specifically, it is proposed that risk attitude, risk perception and numerical ability will mediate the relationship between age and adherence to health guidelines ('COVID-19 risk taking'). While the questions outlined in this

Stage 1 manuscript are timely, I have questions and comments about the rationale, method and analytical approach.

1. There needs to be further justification for the implied causal paths in the model. What, for example is the rationale for numeracy being directly related to risk taking? What is numeracy is only related to risk taking via risk perception?
2. The model assumes that the paths from age to the three mediators is equal in both age groups. How can this assumption be justified?
3. I was surprised to see age as a categorical variable. Why not model age as a continuous variable. What justifies the use of these two age groups and is the use of a categorical independent variable problematic for the mediation analysis?
4. I do not think that averaging the survey items that you believe reflect the mediators and the dependent variable is an optimal approach here. What if what you think are indicators of the same underlying construct turn out not to be? Then your mean scores will not be particularly meaningful. Relatedly, what if there are differences in the reliability of different survey items? To address such potential issues, it would be useful to take a latent variable approach and use SEM.
5. You do not provide any justification for the effect size you used to estimate statistical power. How did you arrive at this effect size? Also, there is no link to the cited supplementary information.
6. The choice of the numeracy measure is not clearly justified. In the introduction you talk about the importance of understanding proportions, percentages, fractions etc. It is not clear whether the 'Objective Numeracy Measure' measures aspects of numerical understanding and skills that are relevant to understanding COVID-19 data. In other words, numeracy may not uniformly relate to adherence to public health advice but instead it may be a subset of numeracy skills (in particular being able to understand, process and calculate with proportions) that are particularly relevant. Are you sure you are measuring aspects of numeracy that have previously been found to be associated with health-related behaviours?
7. The description of the survey items should be more detailed and a link to a document containing all the survey items should to be provided.

Reviewer: 2

Comments to the Author(s)

Wolfe and colleagues propose to investigate mediating factors affecting the relationship between age and Covid-19 related risk-taking behavior. The team has taken great care in presenting the background, hypotheses and especially methodology and data analysis plan. However, the first two elements, the background and hypotheses, need more consideration. The motivation behind the study is not entirely clear, given the hypotheses posed. While there is an urgent need to understand the pandemic's various ramifications, studies need to be grounded in objective reference frames to encourage public trust in research.

1. The scientific validity of the research question(s)

The most problematic issue in this registered report is the mishandling of the term risk. In particular, there is an assumed, implied and unverified relationship between Covid-19 preventative behaviors, risk "aversion" and rationality. However, a considerable debate is ongoing regarding the true, objective (country, age and health-condition adjusted) risk of

infection; hospitalization; and mortality. What is the reference? Where are these objective measures in the report? Without that standard, measuring differences in Covid-19 risk-perception and risk-taking becomes a creative, not scientific, exercise.

Indeed, the problem of risk in relation to Covid-19 is that it remains difficult to compute. How do the authors plan to address this?

Further, a remarkable feature of Covid-19 related risk is in its non-uniform distribution, particularly with regards to age. How will the study disentangle true, individual-specific risk from individual perceived risk?

Here is an example of the flaw in the study's premise:

P 4. line 48-49 "First, risk perception can account for the age differences in risk taking; older adults have a higher risk associated with contracting COVID-19 than younger people." This statement encapsulates the main flaw in the study; risk perception and risk are two different things. They are confounded here. Both risk perception and true risk may differ between age groups.

All in all, I am not entirely convinced by the age factor in the model. More interesting would be the numeracy effect on risk-taking etc.

2. The logic, rationale, and plausibility of the proposed hypotheses

Without addressing the key premise of objective risk, it is difficult to judge the logic of the proposed hypotheses. However, the explicit value of finding a significant relationship between two related variables (eg risk perception and risk taking; and risk attitude and risk-taking) is not obvious.

p. 8 line 52: Why is the relationship between risk perception and risk-taking hypothetical?

p. 8. idem line 59 regarding risk attitude and risk-taking.

3. The soundness and feasibility of the methodology and analysis pipeline (including statistical power analysis where applicable)

The above are detailed and appear sound, however, the hypothesis in p.9. lines 7-12 may be tenuous. Here, you could cast a different model where numeracy predicts risk-taking and perhaps probe the mediating effects of age on that relationship. But this proposition still does not account for the disparate risk of Covid-19 on different age groups. Finally, why is the study constrained to the UK? Is this due to the Ethics approval? If it is not, it would be beneficial to open the study to other countries, particularly given the unique opportunity to study a global, rather than local, state of distress.

4. Whether the clarity and degree of methodological detail would be sufficient to replicate exactly the proposed experimental procedures and analysis pipeline

The methodology is well detailed and replicable. Only one minor concern regarding measures of Covid-19 related risk perception:

p.12, lines 20-24 It is not clear how the items are scaled, or what they are scaled by, before being averaged. Are estimated risks of infection, hospitalization and death combined into one measure per respondent?

5. Whether the authors provide a sufficiently clear and detailed description of the methods to prevent undisclosed flexibility in the experimental procedures or analysis pipeline

The authors methods description are transparent, including both the code and questionnaire, leaving little room for methodological subterfuge.

6. Whether the authors have considered sufficient outcome-neutral conditions (e.g. positive controls) for ensuring that the results obtained are able to test the stated hypotheses

This may remain open to interpretation and may also not be feasible given the comparative rarity of a global pandemic. Indeed, hypotheses 1-3 could be viewed as sanity checks, with the 4th hypothesis yielding the most pertinent results,

Minor comments:

p.1, line 60, this is not a mortality rate as it is not scaled to any standard.

p. 5 lines 23-24: "Prior research shows that older adults perceive more risk in health-related activities than younger adults" Please provide a reference.

p. 5 lines 26-33 Do older adults perceive less risk of infection than do younger people? Or do they believe they have less of an infection risk than younger people do? It is not clear here.

p. 5 line 33-35 Please include some references here "Risk perception is a known predictor of risk behavior and has shown to vary between age groups".

p. 6 lines 35-36 Please remove redundancy here.

p. 8 line 24: In the second hypothesis, hasn't this already been reported in another study above?

Author's Response to Decision Letter for (RSOS-201445.R0)

See Appendix A.

RSOS-201445.R1 (Revision)

Review form: Reviewer 1

Do you have any ethical concerns with this paper?

No

Recommendation?

Accept in principle

Comments to the Author(s)

Thank you for carefully addressing all my prior concerns and revising your manuscript accordingly.

Review form: Reviewer 2

Do you have any ethical concerns with this paper?

No

Recommendation?

Reject

Comments to the Author(s)

The premise of the study remains problematic, as it omits key assumptions, chief among them an understanding of decision-making under risk. Even non-human animals integrate risk (<https://doi.org/10.1007/s00213-012-2854-2>, <https://doi.org/10.1016/j.biopsycho.2009.04.008>) in their decision-making. A central point of the study rests on age-related differences in risk-taking/perception; but if there is one element of Covid19 that is known, it's that age is a risk factor. What if young people take more risks simply because they have learned, over time, that said risk is low enough? What if old people take less risks simply because they have learned, over time that said risk is too high? One possibility is to split groups and examine old and young separately.

Another problem that persists in this report is the casting of true risk as immaterial to the research question. It is fundamental. Without a ground truth – even an approximation of the latter – what is numeracy compared against? The authors implicitly cast government guidelines as counterpoints to true risk, yet do not explicitly say so. If the authors deem governmental guidelines to be adequate responses to the true risk – and therefore cast a lack of compliance with said guidelines as a deviation from accurate risk perception – then that definition needs to be explicitly stated.

“These findings could benefit risk communication during the remainder of the pandemic as well as after, as it highlights what areas communication should focus on.”

How is this possible if the true risk isn't known? What would be communicated?

“For instance, a meta-analysis by Brewer et al. (2007) found that those who perceived higher risk (higher likelihood of getting an illness, susceptibility to the illness, and severity of the illness) were more likely to get vaccinated.”

One could argue that such people may also suffer from hypochondria. We cannot know unless there's a ground truth for real risk or danger. If the UK government's guidelines are the reference, and a deviation from said guidelines is “risk-taking”, then this should be stated.

“People's objective risk is not always reflected in their risk perception. An example of such dissonance was found by Kataponi et al. (2004) in their meta-analysis, in which younger women reported higher risk perception of breast cancer than older women, despite older age being an established risk factor for breast cancer.”

An imperfect correlation is not a dissonance. Further, it is unclear what point is being made here. It seems to be: risk perception has little to do with objective risk; high risk perception predicts risk aversive behavior, which is good, regardless of objective risk. Such logic could just as easily be used to encourage superstitious practices, which are based on risk perception, regardless of objective risk, and, I would wager, more common among older people.

“The disparity between subjective and objective risk is further supported by findings by Brewer and Hallman (2006), who found that participants' risk perception fully mediated the relationship between objective risk and influenza vaccination.”

This does not preclude that risk perception reflects the true risk. From the cited paper "This finding supports the conclusion that objective risk affects vaccination only insofar as it changes subjective risk (i.e., full mediation)[23]. (PDF) Subjective and Objective Risk as Predictors of Influenza Vaccination during the Vaccine Shortage of 2004–2005. Available from: https://www.researchgate.net/publication/6713966_Subjective_and_Objective_Risk_as_Predictors_of_Influenza_Vaccination_during_the_Vaccine_Shortage_of_2004-2005 [accessed Sep 29 2020]."

"As following guidelines is a key to preventing the spread of the virus, risk attitudes could provide more information on why people differ in how strict they adhere to guidelines."

Here is implicitly what is cast as objective risk. This measure needs to be brought to the fore, explicitly.

"The ability to comprehend these numbers and apply them to calculate a useful statistic may influence people's willingness to take risks. Some may find these numbers confusing or difficult and may make miscalculations, which may cause misconception about the virus' severity, and may influence behaviour towards limiting the spread of the virus"

These numbers are putatively presented in order to inform the public – on objective risk. Therefore, the study's hypotheses rest on an assumption that individuals do handle objective risk. This handling of risk is assumed to correlate with numeracy. Thus, there remains a discrepancy in how the hypotheses are formulated. If objective risk is not considered, what is numeric ability associated with?

Decision letter (RSOS-201445.R1)

Dear Ms Wolfe,

The Editors assigned to your revised Stage 1 Registered Report ("Age differences in COVID-19 risk-taking, and the relationship with risk attitude and numerical ability.") have now received comments from reviewers. We would like you to revise your paper in accordance with the referee and editors suggestions which can be found below (not including confidential reports to the Editor). Please note this decision does not guarantee eventual acceptance.

Please submit a copy of your revised paper within three weeks (i.e. by the 27-Oct-2020). If we do not hear from you within this time then it will be assumed that the paper has been withdrawn.

When submitting your revised manuscript, you must respond to the comments made by the referees and upload a file "Response to Referees". Please use this to document how you have responded to the comments, and the adjustments you have made. In order to expedite the processing of the revised manuscript, please be as specific as possible in your response.

on behalf of Professor Chris Chambers (Registered Reports Editor, Royal Society Open Science)
openscience@royalsociety.org

Associate Editor Comments to Author (Professor Chris Chambers):

Associate Editor: 1

Comments to the Author:

The revised Stage 1 manuscript was re-assessed but both reviewers, who give polarised opinions (Accept vs Reject). As you will see, Reviewer 1 is satisfied and recommends in-principle acceptance, while Reviewer 2 remains unconvinced that the study can achieve the intended goal without explicitly taking into account objective risk. I certainly see merit in the reviewer's concerns, and it is clear that the reviewer is dissatisfied with the counter arguments offered in your first response.

In weighing up the reviews, I initially leaned toward rejection as I suspect this may be an unshiftable roadblock and the bar for COVID-19 RRs is necessarily high. However, on reflection I can also see a potential counterargument that there is merit in studying risk perception and risk attitudes in the absence of objective risk provided the aims and conclusions are extremely limited -- and it is possible in this case that the aims of the study are simply too ambitious given the limitations. I admit I am undecided but I want to give you the opportunity to submit a final revision and rebuttal. In revising it is essential that this issue is addressed in the revised manuscript (not just in the response), either through a design revision or through a discussion of the issue in the main text. Based on this, I may then consult with Reviewer 2 (if they are available) and among the editorial board to reach a final decision as soon as possible.

I cannot promise that the outcome of this final process will be a positive one for the authors, and so if you are under a time constraint (and if you strongly disagree with Reviewer 2 and would simply repeat the arguments already made in your first response) then you may prefer to press ahead with the study on your own terms.

Comments to Author:

Reviewer: 1

Comments to the Author(s)

Thank you for carefully addressing all my prior concerns and revising your manuscript accordingly.

Reviewer: 2

Comments to the Author(s)

The premise of the study remains problematic, as it omits key assumptions, chief among them an understanding of decision-making under risk. Even non-human animals integrate risk

(<https://doi.org/10.1007/s00213-012-2854-2>, <https://doi.org/10.1016/j.biopsycho.2009.04.008>) in their decision-making. A central point of the study rests on age-related differences in risk-taking/perception; but if there is one element of Covid19 that is known, it's that age is a risk factor. What if young people take more risks simply because they have learned, over time, that said risk is low enough? What if old people take less risks simply because they have learned, over time that said risk is too high? One possibility is to split groups and examine old and young separately.

Another problem that persists in this report is the casting of true risk as immaterial to the research question. It is fundamental. Without a ground truth – even an approximation of the latter – what is numeracy compared against? The authors implicitly cast government guidelines as counterpoints to true risk, yet do not explicitly say so. If the authors deem governmental guidelines to be adequate responses to the true risk – and therefore cast a lack of compliance with said guidelines as a deviation from accurate risk perception – then that definition needs to be explicitly stated.

“These findings could benefit risk communication during the remainder of the pandemic as well as after, as it highlights what areas communication should focus on.”

How is this possible if the true risk isn't known? What would be communicated?

“For instance, a meta-analysis by Brewer et al. (2007) found that those who perceived higher risk (higher likelihood of getting an illness, susceptibility to the illness, and severity of the illness) were more likely to get vaccinated.”

One could argue that such people may also suffer from hypochondria. We cannot know unless there's a ground truth for real risk or danger. If the UK government's guidelines are the reference, and a deviation from said guidelines is “risk-taking”, then this should be stated.

“People's objective risk is not always reflected in their risk perception. An example of such dissonance was found by Kataponi et al. (2004) in their meta-analysis, in which younger women reported higher risk perception of breast cancer than older women, despite older age being an established risk factor for breast cancer.”

An imperfect correlation is not a dissonance. Further, it is unclear what point is being made here. It seems to be: risk perception has little to do with objective risk; high risk perception predicts risk aversive behavior, which is good, regardless of objective risk. Such logic could just as easily be used to encourage superstitious practices, which are based on risk perception, regardless of objective risk, and, I would wager, more common among older people.

“The disparity between subjective and objective risk is further supported by findings by Brewer and Hallman (2006), who found that participants' risk perception fully mediated the relationship between objective risk and influenza vaccination.”

This does not preclude that risk perception reflects the true risk. From the cited paper

“This finding supports the conclusion that objective risk affects vaccination only insofar as it changes subjective risk (i.e., full mediation)[23].

(PDF) Subjective and Objective Risk as Predictors of Influenza Vaccination during the Vaccine Shortage of 2004–2005. Available from:

https://www.researchgate.net/publication/6713966_Subjective_and_Objective_Risk_as_Predictors_of_Influenza_Vaccination_during_the_Vaccine_Shortage_of_2004-2005 [accessed Sep 29 2020].”

“As following guidelines is a key to preventing the spread of the virus, risk attitudes could provide more information on why people differ in how strict they adhere to guidelines.”

Here is implicitly what is cast as objective risk. This measure needs to be brought to the fore, explicitly.

“The ability to comprehend these numbers and apply them to calculate a useful statistic may influence people’s willingness to take risks. Some may find these numbers confusing or difficult and may make miscalculations, which may cause misconception about the virus’ severity, and may influence behaviour towards limiting the spread of the virus”

These numbers are putatively presented in order to inform the public – on objective risk. Therefore, the study’s hypotheses rest on an assumption that individuals do handle objective risk. This handling of risk is assumed to correlate with numeracy. Thus, there remains a discrepancy in how the hypotheses are formulated. If objective risk is not considered, what is numeric ability associated with?

Author's Response to Decision Letter for (RSOS-201445.R1)

See Appendix B.

RSOS-201445.R2 (Revision)

Review form: Reviewer 2

Do you have any ethical concerns with this paper?

No

Recommendation?

Accept in principle

Comments to the Author(s)

Thank you for addressing my major concerns regarding the design. The study is much improved because of it and will inspire more confidence in its results.

Decision letter (RSOS-201445.R2)

Dear Ms Wolfe

On behalf of the Editor, I am pleased to inform you that your Manuscript RSOS-201445.R2 entitled "Age differences in COVID-19 risk-taking, and the relationship with risk attitude and numerical ability." has been accepted in principle for publication in Royal Society Open Science. The reviewers' and editors' comments are included at the end of this email.

You may now progress to Stage 2 and complete the study as approved. Please note that your approved Stage 1 manuscript has now been publicly registered at <https://doi.org/10.17605/OSF.IO/3NV56>

In due course, please include the following statement in the Stage 2 manuscript: "This study was preregistered prior to data collection and analysis on 6 November 2020. The accepted Stage 1 manuscript, unchanged from the point of in-principle acceptance, may be viewed at <https://doi.org/10.17605/OSF.IO/3NV56>"

Please note that your manuscript can still be rejected for publication at Stage 2 if the Editors consider any of the following conditions to be met:

- The results were unable to test the authors' proposed hypotheses by failing to meet the approved outcome-neutral criteria.
- The authors altered the Introduction, rationale, or hypotheses, as approved in the Stage 1 submission.
- The authors failed to adhere closely to the registered experimental procedures. Please note that any deviations from the approved experimental procedures must be communicated to the editor immediately for approval, and prior to the completion of data collection. Failure to do so can result in revocation of in-principle acceptance and rejection at Stage 2 (see complete guidelines for further information).
- Any post-hoc (unregistered) analyses were either unjustified, insufficiently caveated, or overly dominant in shaping the authors' conclusions.
- The authors' conclusions were not justified given the data obtained.

We encourage you to read the complete guidelines for authors concerning Stage 2 submissions at <https://royalsocietypublishing.org/rsos/registered-reports#ReviewerGuideRegRep>. Please especially note the requirements for data sharing, reporting the URL of the independently registered protocol, and that withdrawing your manuscript will result in publication of a Withdrawn Registration.

Once again, thank you for submitting your manuscript to Royal Society Open Science and we look forward to receiving your Stage 2 submission. If you have any questions at all, please do not hesitate to get in touch. We look forward to hearing from you shortly with the anticipated submission date for your stage two manuscript.

on behalf of Professor Chris Chambers (Associate Editor) and Chris Chambers (Registered Reports Editor, Royal Society Open Science)
openscience@royalsociety.org

Associate Editor Comments to Author (Professor Chris Chambers):
Associate Editor: 1
Comments to the Author:

Reviewer 2 is now satisfied with the manuscript. My thanks to the reviewer for the multiple repeated (rapid) reviews and I am happy to now award Stage 1 IPA.

Reviewers' comments to Author:

Reviewer: 2

Comments to the Author(s)

Thank you for addressing my major concerns regarding the design. The study is much improved because of it and will inspire more confidence in its results.

Author's Response to Decision Letter for (RSOS-201445.R2)

See Appendix C.

RSOS-201445.R3 (Revision)

Review form: Reviewer 2

Is the manuscript scientifically sound in its present form?

No

Are the interpretations and conclusions justified by the results?

No

Is the language acceptable?

No

Do you have any ethical concerns with this paper?

No

Have you any concerns about statistical analyses in this paper?

No

Recommendation?

Major revision

Comments to the Author(s)

The results of this registered report do not confirm the initial hypotheses, namely that low numeracy would predict more "risk-taking" concerning Covid measures. There are two biases used as assumptions in this study that are highly problematic and persist in spite of the data : 1) the notion that "low-numeracy" people adhere less to Covid measures; 2) that the government, and specifically the UK government, provides scientifically accurate guidance on controlling the pandemic.

1. Whether the data are able to test the authors' proposed hypotheses by passing the approved outcome-neutral criteria (such as absence of floor and ceiling effects or success of positive controls)

Yes

2. Whether the Introduction, rationale and stated hypotheses are the same as the approved Stage 1 submission

Yes

3. Whether the authors adhered precisely to the registered experimental procedures

Yes

4. Where applicable, whether any unregistered exploratory statistical analyses are justified, methodologically sound, and informative

The exploratory analyses do not appear justified, methodologically or informative. I advise the authors to, if they have a solid theoretical framework for one of these analyses, to better tighten their hypothesis and justification for it. Otherwise, should perform a follow-up study on separate cohort.

5. Whether the authors' conclusions are justified given the data

Some of the data, especially those concerning numeracy and risk-taking, are very interesting; contra to the authors expectations; and deserve to be highlighted better than they are. On the other hand, the proliferation of qualitative risk measures muddy the picture of how risk affects behavior. This aspect of the paper deserves a more careful and considered approach in the discussion.

Major points:

Concerning the first point: Nowhere in the study – on risk no less – do the authors mention rationality or what constitutes low or high risk. Indeed, from the outset, it should have been clear that young people with high numeracy – that is people who can understand probabilities and percentages, and case fatality rates and so on – would adhere less to Covid measures because the risk of complications due to Covid remains low in the young. From my reading, the young act rationally – computing risk and guiding their actions accordingly. The finding (if read correctly – the specification of directionality in effects is unclear) that older people with low numeracy adhere less to Covid measures would seem to support the initial hypothesis concerning Covid and highlight a potential area to fix so to speak. But generally, the authors should question the very premise that less adherence to Covid measures actually constitutes risk-taking – or is it rational behavior?

Concerning the second point: The UK has one of the highest Covid mortality rates in OECD countries. In addition, its political body would appear to have gone through considerable volatility during the pandemic. As far as this reviewer understands, the UK health ministry is not led by a doctor or scientist. Thus one is left to wonder why a citizen adhering to government mandates would be called risk-averse (in relation to health). In other words, by what measure is the UK government trustworthy? If that question is deemed inappropriate to this review, the authors should reconsider the equating of following UK government mandates with reasonableness/pro-sociality/rationality and/or risk aversion.

While I welcome the addition of exploratory analyses, I feel that in this case they violate the spirit of registered reports. They read as fishing expeditions. All or most should be omitted from the study. The first recommendation concerns the analyses on government ordered vs recommended guidelines. I am not convinced this distinction is pertinent. It is also a tad provincial in that governments worldwide have drawn that distinction differently and therefore

one cannot, based on these results, determine if differences are due to the quality of the guidelines (mandatory vs not) or to the guidelines themselves.

The models concerning risk perception also appear to be very ad-hoc. Given that risk of Covid contraction; complication; and death exist as probabilities, why not ask respondents to state that probability (percentage) to compare to their objective risk? Respondents are left with ambiguous statements to rate. Then with regards to assessing risk to self and others, please refer to Covid-specific studies on the latter:

<https://doi.org/10.1002/ejsp.2737>

[10.31234/osf.io/qrbza](https://osf.io/qrbza)

<https://doi.org/10.1016/j.puhe.2020.06.032>

Finally, given that the results don't confirm initial hypotheses, please: 1) emphasize this more clearly in the discussion; 2) streamline the writing. It is no longer clear exactly, given all the different results, what is interesting and why. There is an opportunity to do so in the discussion especially.

Some more detailed comments:

Language:

Please review the text and streamline the language. There is considerable repetition (e.g. pp.45 and 46 on risk perception for self and others). While I do not agree with word limits, the manuscript is too long and could do with a heavy editing hand.

Please check time indicators (are we still in a second wave, for instance?) and ensure that time-related terms remain coherent for a reader in the far-off future (e.g. make sure to include year, e.g. p.6, line 7).

Please make a table for all the risk measures listed (risk perception, objective risk, risk attitude, etc.) The field is large and nomenclature varies so it would be helpful to keep track of what's what.

References:

p.3 line 37, please provide a reference to the notion that adherence to, specifically, government-mandated directives enhances public health, especially given that you later divide public health measures into government-mandated versus recommended.

p.7 line 57, please provide one or more recent references (the stability of risk preferences have been repeatedly challenged).

Data analysis and results:

p.8, line 55 Please provide directions in effects. What was the effect of numeracy in the Petrova et al. study, positive or negative?

p. 19. line 31, 32 is unclear.

p. 19, final lines: why weren't these variables included in the final analysis? Or, alternatively, why were they included?

Figure 2: Can you add information to the caption or alternatively change or omit the figure? I see very little difference between the age groups.

There remains a methodological issue concerning objective risk and age, It is clear, and confirmed, that these two variables are confounded (this paper lists an r of 0.72, which is on the stunning side in social science) so much so that they are almost the same.

p.27 on numeracy and risk taking between age groups, please provide a test statistic on the numeracy item.

p. 28 lines 49-53 Please specify direction of effects, especially given that you found an effect opposite to the one expected.

Please include effect sizes where relevant.

Decision letter (RSOS-201445.R3)

Dear Mrs Wolfe,

The editors assigned to your Stage 2 Registered Report ("Age differences in COVID-19 risk-taking, and the relationship with risk attitude and numerical ability.") has now received comments from reviewers. We would like you to revise your paper in accordance with the referee and Subject Editor suggestions which can be found below (not including confidential reports to the Editor).

Please submit a copy of your revised paper within three weeks (i.e. by the 24-Aug-2021).

- Data accessibility

If you wish to submit your supporting data or code to Dryad (<http://datadryad.org/>), or modify your current submission to dryad, please use the following link:
<http://datadryad.org/submit?journalID=RSOS&manu=RSOS-201445.R3>

- Competing interests

- Authors' contributions

- Acknowledgements

- Funding statement

Kind regards,
Royal Society Open Science Editorial Office
Royal Society Open Science

on behalf of Chris Chambers

Subject Editor, Royal Society Open Science
 openscience@royalsociety.org

Associate Editor's comments (Professor Chris Chambers):

One of the original Stage 1 reviewers was available to evaluate the Stage 2 manuscript. As you will see, the review is quite critical of several aspects of the final report, primarily the justification for the reported exploratory analyses, the clarity of the Discussion (and, in turn, clarity of the reporting of the conclusions), and the clear separation of confirmatory and exploratory outcomes. I must admit that my own reading of the manuscript was somewhat less negative than this assessment, although I yield to the specialist expertise of the reviewer and agree that the manuscript is relatively heavy" on exploratory analyses. Please respond thoroughly in a revision or rebuttal to the concerns raised and ensure that the confirmatory outcomes play centre stage in the conclusions and abstract.

In revising, please be sure to make no changes to the approved Stage 1 part of the manuscript unless doing so is necessary to correct errors of fact (or other minor errors, such as typographic or grammatical errors).

Reviewers' Comments to Author:

Reviewer: 2

Comments to the Author(s)

The results of this registered report do not confirm the initial hypotheses, namely that low numeracy would predict more "risk-taking" concerning Covid measures. There are two biases used as assumptions in this study that are highly problematic and persist in spite of the data : 1) the notion that "low-numeracy" people adhere less to Covid measures; 2) that the government, and specifically the UK government, provides scientifically accurate guidance on controlling the pandemic.

1. Whether the data are able to test the authors' proposed hypotheses by passing the approved outcome-neutral criteria (such as absence of floor and ceiling effects or success of positive controls)

Yes

2. Whether the Introduction, rationale and stated hypotheses are the same as the approved Stage 1 submission

Yes

3. Whether the authors adhered precisely to the registered experimental procedures

Yes

4. Where applicable, whether any unregistered exploratory statistical analyses are justified, methodologically sound, and informative

The exploratory analyses do not appear justified, methodologically or informative. I advise the authors to, if they have a solid theoretical framework for one of these analyses, to better tighten their hypothesis and justification for it. Otherwise, should perform a follow-up study on separate cohort.

5. Whether the authors' conclusions are justified given the data

Some of the data, especially those concerning numeracy and risk-taking, are very interesting; contra to the authors expectations; and deserve to be highlighted better than they are. On the other hand, the proliferation of qualitative risk measures muddy the picture of how risk affects behavior. This aspect of the paper deserves a more careful and considered approach in the discussion.

Major points:

Concerning the first point: Nowhere in the study – on risk no less – do the authors mention rationality or what constitutes low or high risk. Indeed, from the outset, it should have been clear that young people with high numeracy – that is people who can understand probabilities and percentages, and case fatality rates and so on - would adhere less to Covid measures because the risk of complications due to Covid remains low in the young. From my reading, the young act rationally – computing risk and guiding their actions accordingly. The finding (if read correctly – the specification of directionality in effects is unclear) that older people with low numeracy adhere less to Covid measures would seem to support the initial hypothesis concerning Covid and highlight a potential area to fix so to speak. But generally, the authors should question the very premise that less adherence to Covid measures actually constitutes risk-taking – or is it rational behavior?

Concerning the second point: The UK has one of the highest Covid mortality rates in OECD countries. In addition, its political body would appear to have gone through considerable volatility during the pandemic. As far as this reviewer understands, the UK health ministry is not led by a doctor or scientist. Thus one is left to wonder why a citizen adhering to government mandates would be called risk-averse (in relation to health). In other words, by what measure is the UK government trustworthy? If that question is deemed inappropriate to this review, the authors should reconsider the equating of following UK government mandates with reasonableness/pro-sociality/rationality and/or risk aversion.

While I welcome the addition of exploratory analyses, I feel that in this case they violate the spirit of registered reports. They read as fishing expeditions. All or most should be omitted from the study. The first recommendation concerns the analyses on government ordered vs recommended guidelines. I am not convinced this distinction is pertinent. It is also a tad provincial in that governments worldwide have drawn that distinction differently and therefore one cannot, based on these results, determine if differences are due to the quality of the guidelines (mandatory vs not) or to the guidelines themselves.

The models concerning risk perception also appear to be very ad-hoc. Given that risk of Covid contraction; complication; and death exist as probabilities, why not ask respondents to state that probability (percentage) to compare to their objective risk? Respondents are left with ambiguous statements to rate. Then with regards to assessing risk to self and others, please refer to Covid-specific studies on the latter:

<https://doi.org/10.1002/ejsp.2737>

[10.31234/osf.io/qrbza](https://doi.org/10.31234/osf.io/qrbza)

<https://doi.org/10.1016/j.puhe.2020.06.032>

Finally, given that the results don't confirm initial hypotheses, please: 1) emphasize this more clearly in the discussion; 2) streamline the writing. It is no longer clear exactly, given all the different results, what is interesting and why. There is an opportunity to do so in the discussion especially.

Some more detailed comments:

Language:

Please review the text and streamline the language. There is considerable repetition (e.g. pp.45 and 46 on risk perception for self and others). While I do not agree with word limits, the manuscript is too long and could do with a heavy editing hand.

Please check time indicators (are we still in a second wave, for instance?) and ensure that time-related terms remain coherent for a reader in the far-off future (e.g. make sure to include year, e.g. p.6, line 7).

Please make a table for all the risk measures listed (risk perception, objective risk, risk attitude, etc.) The field is large and nomenclature varies so it would be helpful to keep track of what's what.

References:

p.3 line 37, please provide a reference to the notion that adherence to, specifically, government-mandated directives enhances public health, especially given that you later divide public health measures into government-mandated versus recommended.

p.7 line 57, please provide one or more recent references (the stability of risk preferences have been repeatedly challenged).

Data analysis and results:

p.8, line 55 Please provide directions in effects. What was the effect of numeracy in the Petrova et al. study, positive or negative?

p. 19. line 31, 32 is unclear.

p. 19, final lines: why weren't these variables included in the final analysis? Or, alternatively, why were they included?

Figure 2: Can you add information to the caption or alternatively change or omit the figure? I see very little difference between the age groups.

There remains a methodological issue concerning objective risk and age, It is clear, and confirmed, that these two variables are confounded (this paper lists an r of 0.72, which is on the stunning side in social science) so much so that they are almost the same.

p.27 on numeracy and risk taking between age groups, please provide a test statistic on the numeracy item.

p. 28 lines 49-53 Please specify direction of effects, especially given that you found an effect opposite to the one expected.

Please include effect sizes where relevant.

Author's Response to Decision Letter for (RSOS-201445.R3)

See Appendix D.

Decision letter (RSOS-201445.R4)

Dear Dr Wolfe:

It is a pleasure to accept your manuscript entitled "Age differences in COVID-19 risk-taking, and the relationship with risk attitude and numerical ability" in its current form for publication in Royal Society Open Science.

COVID-19 rapid publication process:

We are taking steps to expedite the publication of research relevant to the pandemic. If you wish, you can opt to have your paper published as soon as it is ready, rather than waiting for it to be published the scheduled Wednesday.

This means your paper will not be included in the weekly media round-up which the Society sends to journalists ahead of publication. However, it will still appear in the COVID-19 Publishing Collection which journalists will be directed to each week (<https://royalsocietypublishing.org/topic/special-collections/novel-coronavirus-outbreak>).

If you wish to have your paper considered for immediate publication, or to discuss further, please notify openscience_proofs@royalsociety.org and press@royalsociety.org when you respond to this email.

on behalf of Professor Chris Chambers (Subject Editor)
openscience@royalsociety.org

Appendix A

Response to reviewer comments

Reviewer 1

Comments to the Author(s)

The aim of this study is to better understand why younger and older adults differ in their adherence to COVID-19 related public health guidelines. Specifically, it is proposed that risk attitude, risk perception and numerical ability will mediate the relationship between age and adherence to health guidelines ('COVID-19 risk taking'). While the questions outlined in this Stage 1 manuscript are timely, I have questions and comments about the rationale, method and analytical approach.

1. There needs to be further justification for the implied causal paths in the model. What, for example is the rationale for numeracy being directly related to risk taking? What is numeracy is only related to risk taking via risk perception?

RESPONSE: Thank you for your helpful feedback. We provided further justification in the manuscript. We decided to include numeracy as a predictor for risk-taking as the communication concerning the current pandemic has largely involved the use of numerical information. For instance, counts of total infections and deaths, graphs detailing the spread of the virus, and more recently, the use of the reproduction number R. This numerical information is provided to the public under the assumption that the majority, if not all, citizens understand what these numbers mean and represent. We believe that the (mis)understanding of numerical information may (partly) drive health-behaviours during the pandemic.

There are several prior studies that have found a relationship between numeracy and risk-taking, some specifically concerning health-related risk-taking. Petrova et al. (2017) found that participants with low numerical ability were four times more likely to delay critically needed medical treatment. The authors state that the effect of numeracy was a unique predictor to longer decision delays (i.e. time between symptom onset to decision to seek medical care), leading to significant increase in risk for death and serious disability. Låg et al. (2014) reported that numeracy was the strongest predictor of performance on several risk estimation and probability judgments tasks relevant to interpreting medical data and estimating health risk, even after controlling for several covariates. Leiter et al. (2018) found that those with low numeracy skills made worse patient prognostic estimates (participants were given case studies), as well as selecting treatments ill-fitting with patient prognosis (e.g. selecting an aggressive treatment for a 90-year old man with 0% chance of survival or functional independence). Yamashita et al. (2018) investigated numeracy and preventative health behaviours and found that low numerical ability was associated with lower likelihood of dental check-ups in older adults. Additionally, Peters et al. (2014) found that lower numerical ability was associated with a lower willingness to take medication (participants were asked to calculate the likelihood of severe side effects prior to this, with information provided to them).

Due to these findings and the reliance on numerical information during this pandemic, we believe that there is a relationship between someone's numerical ability and their likelihood of adopting preventive health behaviours during the pandemic (not adopting preventive health behaviours is being considered as risk-taking, as one exposes themselves and/or others to possible harm). These studies suggest that numerical ability can be a direct predictor of (high and low risk) health behaviours, and we are interested in assessing whether this also applies to the current project.

We value your insightful suggestion that risk perception may mediate the relationship between numeracy and risk-taking instead. We will include the suggested analysis in the exploratory analysis section of the proposed project.

2. The model assumes that the paths from age to the three mediators is equal in both age groups. How can this assumption be justified?

RESPONSE: Thank you for directing our attention to this potential issue. If we understood your comment correctly, this should not be an issue: age (younger vs older adults) is our independent variable in the model and, therefore, its effect (the path from the independent variable to the mediators) cannot be moderated by the same variable. We apologise if this was not clear in our manuscript since we used both terms “age” and “age groups” in our manuscript. These are equivalent terms in our research. In the method section, when we discussed age, we intended the two age groups that we plan to recruit for this study; younger adults aged between 18 and 35, and older adults aged 65 and older. As we realize that this may have been confusing, we have changed this in the manuscript to make clear that we mean these two age groups for the proposed study, not age as a continuous variable. We hope this addresses your concern but please let us know if we misconstrued your concern.

3. I was surprised to see age as a categorical variable. Why not model age as a continuous variable. What justifies the use of these two age groups and is the use of a categorical independent variable problematic for the mediation analysis?

RESPONSE: Thank you for your valuable suggestion. The reason why we are not treating age as a continuous variable is because we are sampling from the young adults (target $n = 200$, age: 18 - 35) and old adults (target $n = 200$, age: 65 and older). Such a variable cannot be treated as a continuous, normally distributed, variable. There are three additional reasons why age is a categorical variable in this proposed project. First, the proposed sampling plan allows us to maximise the statistical power by focusing on the extreme groups, a common approach in age research. Secondly, using separate groups is frequently done in research on age and risk-taking (e.g., Bonem et al., 2015; Daekin et al., 2004; Denburg et al., 2005; Lauriola & Levin, 2001), and often with the same, or a similar, age range (e.g., Baena et al., 2010; Mather et al., 2009; Pachur et al., 2017; Rolison et al., 2017; Weller et al., 2011). This allows a comparison between the findings of this study to those done prior. We agree that future research should sample age as a continuous variable to better map the developmental trajectories of risk-taking. Third, these specific age groups were chosen due to the issue currently being addressed. The British Secretary of State for Health and Social Care recently stated that “the rise in cases in Bolton is partly due to socialising by people in their 20s and 30s, we know this from contact tracing”. As this group is largely seen as the group responsible for the recent rise in cases (as well as generally considered to be risk-taking) there is good reason to test this group versus the group that is generally considered to be most careful, which is the older adult age group. Within age and risk research, these groups are often of interest due to the underlying general belief that young people are risk-takers, while older adults prefer avoiding risk. Studies on self-reports and behavioural tasks have shown that this is dependent on various factors (e.g. Mata et al., 2011; Frey et al., 2017). This project would contribute as it measures differences in risk-taking between these two groups during a collective experience of risk, something which does not occur often. Including age as a categorical binary variable should not negatively impact the mediation analysis; “if the independent variable, X , is binary, but M and Y are continuous, the standard techniques (in Eqs. (1)–(4)) are perfectly suitable, because X functions only as a predictor variable in the regression equations (Iacobucci, 2012, p. 583).”

4. I do not think that averaging the survey items that you believe reflect the mediators and the dependent variable is an optimal approach here. What if what you think are indicators of the same underlying construct turn out not to be? Then your mean scores will not be particularly meaningful. Relatedly, what if there are differences in the reliability of different survey items? To address such potential issues, it would be useful to take a latent variable approach and use SEM.

RESPONSE: Thank you for your suggestion of using SEM, we agree that this would be a useful approach. However, adopting the use of SEM instead of mediation analysis would require redoing the power analysis. However, we don't currently have all necessary information to rerun this power analysis and there is not enough guidance on how to actually conduct such a power analysis (Westland, 2010). Additionally, by rule of thumb, SEM would likely require a larger sample size for this project to meet the requirement of sufficient power. Unfortunately the sample size, more specifically the monetary amount available to pay participants, is a limitation of the proposed project, and mediation analysis would suit to investigate the questions we are interested in, requiring less participants to have sufficient power.

Saying that, we would like to reassure the reviewer that we tackled the issue raised in this comment.

First, we changed the measures so they would satisfy the criterion of internal consistency that is comparable between the measures (around 0.70-0.80). At this time, almost all of the measures in the proposed project are peer-reviewed (with the exception of the items used to measure risk-taking), and have sufficient reliability:

- The reliability of the Objective Numeracy Scale (Lipkus et al., 2001), including the additional 3 items by Schwartz et al. (1997), was found by Lipkus et al. (2001) to be $\alpha = 0.78$. Weller et al. (2013) reported a Cronbach's alpha of $\alpha = .76$, and Thomson and Oppenheimer (2016) found a Cronbach's alpha of $\alpha = .72$
- The COVID-19 Risk Perception Scale is based on a prior measure developed to measure people's risk perception concerning climate change, and was tested across various countries. The pooled Cronbach's alpha across countries was $\alpha = .72$, the α for the United Kingdom was $\alpha = .80$.
- The internal consistency estimates (i.e., Cronbach's alpha) associated with the 30-item DOSPERT risk-taking scores ranged from $\alpha = .71$ to $\alpha = .86$, and those associated with the risk-perception scores, from $\alpha = .74$ to $\alpha = .83$ (Blais & Weber, 2006).

The items intended to measure risk-taking are the exception to this, and these will be looked at closer once data is available. The dependent variable, preventative health behaviours interpreted as risk-taking, is currently the only variable with a measure that hasn't been tested for its reliability prior to use in this proposed project.

Second, we postulate explicit rules of item exclusion and which individual items will be used as indicators of the variables of interest if the requirements of internal consistency are not met (we will use Cronbach's alpha with a clear cut-off of 0.7).

Finally, other studies with a similar interest have used mediation analysis. For example, Shook et al. (2019) used mediation analysis to assess whether mindfulness mediated the relationship between age and health- and safety risk-taking, also using age as a categorical variable (younger adults were aged 25–36 years, and older were aged 60+ years).

5. You do not provide any justification for the effect size you used to estimate statistical power. How did you arrive at this effect size? Also, there is no link to the cited supplementary information.

RESPONSE: Our apologies for the missing supplementary information, something seems to have gone wrong with the uploaded files. We have included the supplementary materials to

our response, as well as made them available at <https://osf.io/n5y8p/>. We hope it is easily accessible now.

The effect size for numeracy (-0.3) was calculated by us, based on a database with numeracy scores obtained via Prolific Academic for both age groups. This dataset was provided to us by John Kerr and Alex Freeman from the Winton Centre for Risk and Evidence Communication, who are currently doing an extensive investigation of numeracy levels across sampling platforms including Prolific. Based on our prior work, we'd expect the other effects to be larger than that of numeracy, but set everything to the lowest expected effect size in order to be conservative. The other effect sizes, such as age differences in risk-taking (-0.3), risk perception (0.3), and risk attitude (-0.3), are based on the effect size for numeracy.

6. The choice of the numeracy measure is not clearly justified. In the introduction you talk about the importance of understanding proportions, percentages, fractions etc. It is not clear whether the 'Objective Numeracy Measure' measures aspects of numerical understanding and skills that are relevant to understanding COVID-19 data. In other words, numeracy may not uniformly relate to adherence to public health advice but instead it may be a subset of numeracy skills (in particular being able to understand, process and calculate with proportions) that are particularly relevant. Are you sure you are measuring aspects of numeracy that have previously been found to be associated with health-related behaviours?

RESPONSE: Thank you for your comment. We aim to measure participants' general numerical ability as it fits better our rationale. The Objective Numeracy Scale (ONS) can measure general numerical ability as well as subsets of numerical skills (e.g. calculating with proportions and percentages), due to the variation in items. Additionally, the ONS has been used before to measure the relationship between numeracy and health behaviours.

The ONS has been offered in various lengths, as a shorter (3 items) as well as a longer scale (14 items). The studies using subsets of the ONS are discussed below. Låg et al. (2014) is an example of a study who have used both the 11-item version and the 14-item extended version, and found that similar results were derived using both versions of the ONS. They also found that individuals who scored higher on numeracy were less inclined to overestimate risks, reasoned correctly about death rates on the medical data interpretation task, and made fewer errors on a medical-trade-off problem.

Smith et al., (2015) found that those with higher scores performed better on a health task that required them to monitor blood sugar levels, manage prescription medication, and recall asthma symptom prevention triggers. Additionally, Hanoch et al. (2010) found that participants with low scores on the ONS chose the correct interpretation of long-term breast cancer risk less often than those with higher numeracy scores.

Prior studies have used subsets of ONS items to measure participants' numeracy instead; for example, López-Pérez et al. (2017) used four items of the ONS, and found that those who had lower numeracy scores were more likely to choose a radical prostatectomy as a treatment for prostate cancer, compared to those with higher numeracy scores. Choosing radical prostatectomy as a treatment (versus active surveillance, the other option with an equal survival rate) was also associated with the propensity to engage in risky gambling behaviours. Also, Morris et al. (2013) found that adults with low health literacy were more likely to report avoiding doctor's visits, had more fatalistic attitudes toward cancer, and were less accurate in identifying the purpose of cancer screening tests.

The ONS generally involves manipulating and transforming probabilities and proportions, and several items involve scenarios on the likelihood of infection and getting a disease, which is applicable to the current situation with COVID-19. Some examples of items included: "Which of the following numbers represents the biggest risk of getting a disease?", "If Person A's risk of getting a disease is 1% in ten years, and Person B's risk is double that of A's, what is B's risk?", "The chance of getting a viral infection is .0005. Out of 10,000 people, how many of them are expected to get infected?" Because of this, we believe it would be suitable to use in combination with self-reported health behaviours during the

current pandemic. Additionally, if there is interest in separating the items that concern calculating the likelihood of infection or disease, these can be extracted and analysed separately in the exploratory section of the study's analysis.

7. The description of the survey items should be more detailed and a link to a document containing all the survey items should be provided.

RESPONSE: Our apologies, this seems to have been a similar issue to the document with the power analysis. We have attached the supplementary documents, including the survey items, to our resubmission, as well as making them available at <https://osf.io/n5y8p/>.

Reviewer 2

1. The scientific validity of the research question(s)

The most problematic issue in this registered report is the mishandling of the term risk. In particular, there is an assumed, implied and unverified relationship between Covid-19 preventative behaviors, risk “aversion” and rationality. However, a considerable debate is ongoing regarding the true, objective (country, age and health-condition adjusted) risk of infection; hospitalization; and mortality. What is the reference? Where are these objective measures in the report? Without that standard, measuring differences in Covid-19 risk-perception and risk-taking becomes a creative, not scientific, exercise.

Indeed, the problem of risk in relation to Covid-19 is that it remains difficult to compute. How do the authors plan to address this?

Further, a remarkable feature of Covid-19 related risk is in its non-uniform distribution, particularly with regards to age. How will the study disentangle true, individual-specific risk from individual perceived risk?

Here is an example of the flaw in the study’s premise:

P 4. line 48-49 “First, risk perception can account for the age differences in risk taking; older adults have a higher risk associated with contracting COVID-19 than younger people.” This statement encapsulates the main flaw in the study; risk perception and risk are two different things. They are confounded here. Both risk perception and true risk may differ between age groups.

RESPONSE: Firstly, thank you for your feedback and your helpful suggestion. We apologize for the confusion, as we realise that we have not been as clear in the manuscript as we aimed to be. In the proposed project, we do not measure objective risk but are instead interested in people’s subjective (perception of their) risk, and how this affects their preventative behaviour concerning COVID-19. We agree that objective risk is important, and that it differs between age groups, as well as being dependent on other factors such as existing illnesses, gender, country et cetera. Since January this year, our knowledge of the virus has developed and improved. However, we are still limited in our knowledge of risk factors for COVID-19, and so our measurement of an individual’s objective risk would be as well. We also would not have access to medical data that would be needed to make an as-close-as-possible calculation of individual objective risk and would have to rely on participants’ knowledge of their own health and their medical records, to provide information on this. This may not be possible for all participants and risk factors (e.g. their levels of vitamin D, which has been shown to play a role in susceptibility for serious consequences from COVID-19 and are often deficient in those living in the North of England and Scotland). Additionally, there is evidence that people’s objective risk is not always reflected in their risk perception, which might even be true in situations such as COVID-19, in which age groups clearly differ in their objective risk. An example of such dissonance was found by Kataponi et al. (2004) in their meta-analysis, in which younger women reported higher risk perception of breast cancer than older women, despite older age being an established risk factor for breast cancer. However, participants that reported higher risk perception were more likely to adhere to mammography guidelines, as well as being more likely to pursue genetic testing for mutations or undergoing prophylactic mastectomy. This is further supported by findings by Brewer and Hallman (2006), who found that subjective risk (i.e. participants’ perception of being high risk) fully mediated the relationship between objective risk and influenza vaccination. Findings such as these demonstrate the effect of people’s subjective perception of risk on risk-taking behaviours, regardless of their objective risk. Additionally, other studies have found that risk perception is related to risk behaviour. In

terms of health and safety-related risk-taking, a meta-analysis by Brewer et al. (2007) found that those who perceived higher risk (higher likelihood of getting an illness, susceptibility to the illness, and severity of the illness) were more likely to get vaccinated. Hanoch et al. (2018) found that that higher risk perception was associated with a reduced likelihood of engaging in medical risk-taking, and Rhodes and Pivik (2011) found that risk perception was an independent predictor of risky driving behaviour.

Studies focusing on risk perception and adopting protective measures during a pandemic show similar findings; only risk perception was associated with the intent to adopt protective measures during the influenza A (H1N1) pandemic in the Netherlands (Van der Weert, 2011), as well risk perception predicting the adoption of preventive behaviours as in the current pandemic (Bruine de Bruin & Bennett, 2020).

In response to your comment, we have made changes to the manuscript to make it clearer that we are measuring participants' subjective risk, and not objective risk. We also explain why we decided on this approach (similar to the section above). We hope this may clear up any confusion about the aim of the project, and the variables that we aim to measure.

All in all, I am not entirely convinced by the age factor in the model. More interesting would be the numeracy effect on risk-taking etc.

RESPONSE: As part of the mediation model, we are considering the effect of numeracy on risk-taking, this is one of the H1 hypotheses. This relationship is measured as a requirement for numeracy to be a mediator of the relationship between age group and risk-taking. Prior studies have found that older and younger adults often differ in risk-taking, but the direction of this difference, or whether there is a difference at all, is highly dependent on context. By using age as a predictor, and numeracy, risk perception and risk attitude as mediators, we can analyse why older and younger adults differ in their approach. Additionally, a mediation model with numeracy as a predictor, risk perception as a mediator, and risk-taking as a dependent, was suggested by reviewer 1, and we will include this in our exploratory analyses, as this is something we are interested in. Perhaps this fits your expressed interest, as this may provide more information on the effect of numeracy on risk-taking.

2. The logic, rationale, and plausibility of the proposed hypotheses

Without addressing the key premise of objective risk, it is difficult to judge the logic of the proposed hypotheses. However, the explicit value of finding a significant relationship between two related variables (eg risk perception and risk taking; and risk attitude and risk-taking) is not obvious.

RESPONSE: We agree with the reviewer that the value of such testing is not obvious since there are many studies that showed a relationship between risk perception and risk-taking (e.g. Hanoch et al., 2018; Otani & Leonard, 1992; Rhodes and Pivik, 2011). We included this hypothesis for two reasons. First, risk perception and risk-taking are measured in a new context of COVID-19 related perception and behaviour employing new, or less used, measures. Second, confirming the relationship between risk perception and risk-taking is a prerequisite for testing mediation and all hypotheses from H1 set are necessary to be confirmed in our decision tree whether to proceed with the mediation hypotheses (H2 set). To make this more obvious to our readers we now added this rationale to the manuscript.

p.8 line 52: Why is the relationship between risk perception and risk-taking hypothetical?

p.8. idem line 59 regarding risk attitude and risk-taking.

RESPONSE: We agree with the reviewer that you would expect to find a relationship between risk perception and risk-taking, and risk attitude and risk-taking. Some of the measures that we are suggesting for the proposed project are new or have not been applied

often, as well as the situation surrounding COVID-19 being unprecedented. Additionally, we included these hypotheses (H1) as these are a prerequisite for performing a mediation analysis, in which the mediator needs to be significantly related to the dependent variable, and the independent variable. These hypotheses need to be confirmed before we can continue to the main part of our analysis, which involves the H2 hypotheses.

3. The soundness and feasibility of the methodology and analysis pipeline (including statistical power analysis where applicable)

The above are detailed and appear sound, however, the hypothesis in p.9. lines 7- 12 may be tenuous. Here, you could cast a different model where numeracy predicts risk-taking and perhaps probe the mediating effects of age on that relationship.

RESPONSE: Thank you for the suggestion. Age itself cannot be a mediator in this project because age is an independent variable in our model. However, the relationship of age and risk-taking can be (in part) due to age-related differences in numeracy.

But this proposition still does not account for the disparate risk of Covid-19 on different age groups.

RESPONSE: Measuring participants' objective risk for COVID-19 is unfortunately beyond the scope of this study, as well as objective risk often misaligning with people's (subjective) perception of risk (e.g. Brewer and Hallman, 2006; Kataponi et al., 2004). This may even be the case in situations such as COVID-19, in which there are distinct differences in objective risk between age groups. Despite this, subjective perceptions of risk have been associated with risk-taking, and more specially, adopting preventative health measures to avoid risk (Brewer and Hallman, 2006; Brewer et al., 2007; Bruine de Bruin & Bennett, 2020; Hanoch et al., 2018; Kataponi et al., 2004; Van der Weert, 2011). Due to this, we aim to use participants' perception of their risk as a mediator, instead of their objective risk, for COVID-19.

Finally, why is the study constrained to the UK? Is this due to the Ethics approval? If it is not, it would be beneficial to open the study to other countries, particularly given the unique opportunity to study a global, rather than local, state of distress.

RESPONSE: Thank you for this great suggestion. We decided to only run the study in the UK due to the nature of the risk-taking items, which are in line with current government guidelines. Other countries have different guidelines in place for their citizens, as well as different rates of infection and COVID-related deaths, which may affect participants' responses on the measures of this project, such as risk perception and risk-taking. For instance, current government guidelines state that one cannot meet with more than 6 people. This guideline is specifically applicable to the UK and is included as an item in our scale. Locally, not adhering to this guideline can be considered risky as one goes against measures designed to keep them and others safe. However, citizens from other countries can respond that they have met with more than 6 people, but whether that behaviour is risky or not is dependent on their country, the country's infection and death rate, and its approach to COVID-19. To make the data more easily interpretable, we decided to limit the project and its sample to the UK.

Secondly, running the study in multiple countries would require someone able to translate items into the language of that nation, as well as good knowledge of those nation's guidelines. It would also require additional funds to pay participants with. Unfortunately, this is outside the scope of the proposed project. However, this is something we are really interested in, and would be keen to pursue this after the proposed project.

4. Whether the clarity and degree of methodological detail would be sufficient to replicate exactly the proposed experimental procedures and analysis pipeline

The methodology is well detailed and replicable. Only one minor concern regarding measures of Covid-19 related risk perception:

p.12, lines 20-24 It is not clear how the items are scaled, or what they are scaled by, before being averaged. Are estimated risks of infection, hospitalization and death combined into one measure per respondent?

RESPONSE: Indeed, we tried to fix this problem by including a measure that was validated in the prior research. Since submission, new peer-reviewed measures of risk perception have been made available. During the revision period we have taken the opportunity to change items to fit the current situation, including items concerning risk perception. We believe that changing the items for risk perception addresses your concerns listed above and are a better fit for what we are intending to measure. The updated items can be found in supplementary materials 2, and at <https://osf.io/n5y8p/>.

5. Whether the authors provide a sufficiently clear and detailed description of the methods to prevent undisclosed flexibility in the experimental procedures or analysis pipeline

The authors methods description are transparent, including both the code and questionnaire, leaving little room for methodological subterfuge.

6. Whether the authors have considered sufficient outcome-neutral conditions (e.g. positive controls) for ensuring that the results obtained are able to test the stated hypotheses

This may remain open to interpretation and may also not be feasible given the comparative rarity of a global pandemic. Indeed, hypotheses 1-3 could be viewed as sanity checks, with the 4th hypothesis yielding the most pertinent results,

Minor comments:

p.1, line 60, this is not a mortality rate as it is not scaled to any standard.

RESPONSE: Thank you for letting us know, we misused the word mortality rate and have replaced it with “global number of deaths” instead.

p.5 lines 23-24: “Prior research shows that older adults perceive more risk in health- related activities than younger adults” Please provide a reference.

RESPONSE: This section has been rewritten to include a reference, thank you for the feedback.

p.5 lines 26-33 Do older adults perceive less risk of infection than do younger people? Or do they believe they have less of an infection risk than younger people do? It is not clear here.

RESPONSE: This sentence in the manuscript has been rewritten, we hope this clarifies the study’s findings.

p.5 line 33-35 Please include some references here “Risk perception is a known predictor of risk behavior and has shown to vary between age groups”.

RESPONSE: Thank you for the feedback. The paragraph on risk perception has largely been rewritten and this sentence has been removed in that process.

p.6 lines 35-36 Please remove redundancy here.

RESPONSE: The text has been removed from the manuscript per suggestion of the reviewer.

p.8 line 24: In the second hypothesis, hasn't this already been reported in another study above?

RESPONSE: Yes, a recent study by Bruine de Bruin and Bennett (2020) found that risk perception was related to adopting preventative health measures. Our H1: risk perception is a prerequisite for the H2 hypotheses, which are the main part of the mediation analyses. In order to proceed to H2, our main analysis (the mediation model), the H1 hypotheses need to be confirmed first (i.e. we need to establish that there is a relationship between risk perception and risk-taking in our study in order to perform a mediation analysis). If risk perception is not related to risk-taking, it cannot be used as a mediator in the analysis.

References

- Baena, E., Allen, P. A., Kaut, K. P., & Hall, R. J. (2010). On age differences in prefrontal function: the importance of emotional/cognitive integration. *Neuropsychologia*, 48(1), 319–333. <https://doi.org/10.1016/j.neuropsychologia.2009.09.021>
- Blais, A.-R., & Weber, E. U. (2006). A Domain-Specific Risk-Taking (DOSPERT) scale for adult populations. *Judgment and Decision Making*, 1(1), 15.
- Bonem, E. M., Ellsworth, P. C., & Gonzalez, R. (2015). Age Differences in Risk: Perceptions, Intentions and Domains: Age Differences in Risk Taking. *Journal of Behavioral Decision Making*, 28(4), 317–330. <https://doi.org/10.1002/bdm.1848>
- Brewer, N. T., Chapman, G. B., Gibbons, F. X., Gerrard, M., McCaul, K. D., & Weinstein, N. D. (2007). Meta-analysis of the relationship between risk perception and health behavior: The example of vaccination. *Health Psychology*, 26(2), 136–145. <https://doi.org/10.1037/0278-6133.26.2.136>
- Brewer, N. T., & Hallman, W. K. (2006). Subjective and objective risk as predictors of influenza vaccination during the vaccine shortage of 2004-2005. *Clinical Infectious Diseases: An Official Publication of the Infectious Diseases Society of America*, 43(11), 1379–1386. <https://doi.org/10.1086/508466>
- Bruine de Bruin, W., & Bennett, D. (2020). Relationships Between Initial COVID-19 Risk Perceptions and Protective Health Behaviors: A National Survey. *American Journal of Preventive Medicine*, S0749379720302130. <https://doi.org/10.1016/j.amepre.2020.05.001>
- Christopher Westland, J. (2010). Lower bounds on sample size in structural equation modeling. *Electronic Commerce Research and Applications*, 9(6), 476–487. <https://doi.org/10.1016/j.elerap.2010.07.003>
- Deakin, J., Aitken, M., Robbins, T., & Sahakian, B. J. (2004). Risk taking during decision-making in normal volunteers changes with age. *Journal of the International Neuropsychological Society : JINS*, 10(4), 590–598. <https://doi.org/10.1017/S1355617704104104>
- Denburg, N. L., Tranel, D., & Bechara, A. (2005). The ability to decide advantageously declines prematurely in some normal older persons. *Neuropsychologia*, 43(7), 1099–1106. <https://doi.org/10.1016/j.neuropsychologia.2004.09.012>
- Dryhurst, S., Schneider, C. R., Kerr, J., Freeman, A. L. J., Recchia, G., van der Bles, A. M., Spiegelhalter, D., & van der Linden, S. (2020). Risk perceptions of COVID-19 around the world. *Journal of Risk Research*, 1–13. <https://doi.org/10.1080/13669877.2020.1758193>
- Frey, R., Pedroni, A., Mata, R., Rieskamp, J., & Hertwig, R. (2017). Risk preference shares the psychometric structure of major psychological traits. *Science Advances*, 3(10), e1701381. <https://doi.org/10.1126/sciadv.1701381>
- Hanoch, Y., Miron-Shatz, T., & Himmelstein, M. (2010). Genetic testing and risk interpretation: How do women understand lifetime risk results? *Judgment and Decision Making*, 5(2), 8.
- Hanoch, Y., Rolison, J. J., & Freund, A. M. (2018). Does Medical Risk Perception and Risk Taking Change with Age?: Medical Risk Perception and Risk Taking. *Risk Analysis*, 38(5), 917–928. <https://doi.org/10.1111/risa.12692>
- Iacobucci, D. (2012). Mediation analysis and categorical variables: The final frontier. *Journal of Consumer Psychology*, 22(4), 582–594. <https://doi.org/10.1016/j.jcps.2012.03.006>
- Katapodi, M. C., Lee, K. A., Facione, N. C., & Dodd, M. J. (2004). Predictors of perceived breast cancer risk and the relation between perceived risk and breast cancer screening: A meta-analytic review. *Preventive Medicine*, 38(4), 388–402. <https://doi.org/10.1016/j.ypmed.2003.11.012>
- Låg, T., Bauger, L., Lindberg, M., & Friberg, O. (2014). The Role of Numeracy and Intelligence in Health-Risk Estimation and Medical Data Interpretation. *Journal of Behavioral Decision Making*, 27(2), 95–108. <https://doi.org/10.1002/bdm.1788>
- Lauriola, M., & Levin, I. P. (2001). Personality traits and risky decision-making in a controlled

- experimental task: An exploratory study. *Personality and Individual Differences*, 31(2), 215–226. [https://doi.org/10.1016/S0191-8869\(00\)00130-6](https://doi.org/10.1016/S0191-8869(00)00130-6)
- Leiter, N., Motta, M., Reed, R. M., Adeyeye, T., Wiegand, D. L., Shah, N. G., Verceles, A. C., & Netzer, G. (2018). Numeracy and Interpretation of Prognostic Estimates in Intracerebral Hemorrhage among Surrogate Decision Makers in the Neurologic Intensive Care Unit. *Critical Care Medicine*, 46(2), 264–271. <https://doi.org/10.1097/CCM.0000000000002887>
- Lipkus, I. M., Samsa, G., & Rimer, B. K. (2001). General Performance on a Numeracy Scale among Highly Educated Samples. *Medical Decision Making*, 21(1), 37–44. <https://doi.org/10.1177/0272989X0102100105>
- López-Pérez, B., Barnes, A., Frosch, D. L., & Hanoch, Y. (2017). Predicting prostate cancer treatment choices: The role of numeracy, time discounting, and risk attitudes. *Journal of Health Psychology*, 22(6), 788–797. <https://doi.org/10.1177/1359105315615931>
- Mata, R., Josef, A. K., Samanez-Larkin, G. R., & Hertwig, R. (2011). Age differences in risky choice: A meta-analysis. *Annals of the New York Academy of Sciences*, 1235, 18–29. <https://doi.org/10.1111/j.1749-6632.2011.06200.x>
- Mather, M., Gorlick, M. A., & Lighthall, N. R. (2009). To Brake or Accelerate When the Light Turns Yellow?: Stress Reduces Older Adults' Risk Taking in a Driving Game. *Psychological Science*, 20(2), 174–176. <https://doi.org/10.1111/j.1467-9280.2009.02275.x>
- Morris, N. S., Field, T. S., Wagner, J. L., Cutrona, S. L., Roblin, D. W., Gaglio, B., Williams, A. E., Han, P. J. K., Costanza, M. E., & Mazor, K. M. (2013). The Association . Between Health Literacy and Cancer-Related Attitudes, Behaviors, and Knowledge. *Journal of Health Communication*, 18(sup1), 223–241. <https://doi.org/10.1080/10810730.2013.825667>
- Otani, H., Leonard, S. D., Ashford, V. L., Bushroe, M., & Reeder, G. (1992). Age Differences in Perception of Risk. *Perceptual and Motor Skills*, 74(2), 587–594. <https://doi.org/10.2466/pms.1992.74.2.587>
- Pachur, T., Mata, R., & Hertwig, R. (2017). Who Dares, Who Errs? Disentangling Cognitive and Motivational Roots of Age Differences in Decisions Under Risk. *Psychological Science*, 28(4), 504–518. <https://doi.org/10.1177/0956797616687729>
- Peters, E., Hart, P. S., Tusler, M., & Fraenkel, L. (2014). Numbers matter to informed patient choices: A randomized design across age and numeracy levels. *Medical Decision Making : An International Journal of the Society for Medical Decision Making*, 34(4), 430–442. <https://doi.org/10.1177/0272989X13511705>
- Petrova, D., Garcia-Retamero, R., Catena, A., Cokely, E., Heredia Carrasco, A., Arrebola Moreno, A., & Ramírez Hernández, J. A. (2017). Numeracy Predicts Risk of Pre-Hospital Decision Delay: A Retrospective Study of Acute Coronary Syndrome Survival. *Annals of Behavioral Medicine*, 51(2), 292–306. <https://doi.org/10.1007/s12160-016-9853-1>
- Rhodes, N., & Pivik, K. (2011). Age and gender differences in risky driving: The roles of positive affect and risk perception. *Accident Analysis & Prevention*, 43(3), 923–931. <https://doi.org/10.1016/j.aap.2010.11.015>
- Rolison, J. J., Wood, S., & Hanoch, Y. (2017). Age and Adaptation: Stronger Decision Updating about Real World Risks in Older Age. *Risk Analysis*, 37(9), 1632–1643. <https://doi.org/10.1111/risa.12710>
- Shook, N. J., Delaney, R. K., Strough, J., Wilson, J. M., Sevi, B., & Altman, N. (2019). Playing it safe: Dispositional mindfulness partially accounts for age differences in health and safety risk-taking propensity. *Current Psychology*. <https://doi.org/10.1007/s12144-019-0137-3>
- Smith, S. G., Curtis, L. M., O'Connor, R., Federman, A. D., & Wolf, M. S. (2015). ABCs or 123s? The independent contributions of literacy and numeracy skills on health task performance among older adults. *Patient Education and Counseling*, 98(8), 991–997. <https://doi.org/10.1016/j.pec.2015.04.007>
- Thomson, K. S., & Oppenheimer, D. M. (2016). Investigating an alternate form of the

- cognitive reflection test. *Judgment and Decision Making*, 11(1), 15.
- van der Weerd, W., Timmermans, D. R., Beaujean, D. J., Oudhoff, J., & van Steenbergen, J. E. (2011). Monitoring the level of government trust, risk perception and intention of the general public to adopt protective measures during the influenza A (H1N1) pandemic in the Netherlands. *BMC Public Health*, 11(1), 575. <https://doi.org/10.1186/1471-2458-11-575>
- Weller, J. A., Dieckmann, N. F., Tusler, M., Mertz, C. K., Burns, W. J., & Peters, E. (2013). Development and Testing of an Abbreviated Numeracy Scale: A Rasch Analysis Approach. *Journal of Behavioral Decision Making*, 26(2), 198–212. <https://doi.org/10.1002/bdm.1751>
- Weller, J. A., Levin, I. P., & Denburg, N. L. (2011). Trajectory of risky decision making for potential gains and losses from ages 5 to 85. *Journal of Behavioral Decision Making*, 24(4), 331–344. <https://doi.org/10.1002/bdm.690>
- Yamashita, T., Bardo, A. R., Millar, R. J., & Liu, D. (2018). Numeracy and Preventive Health Care Service Utilization among Middle-Aged and Older Adults in the U.S. *Clinical Gerontologist*, 1–12. <https://doi.org/10.1080/07317115.2018.1468378>

Appendix B

Dear Reviewer 2,

Thank you for your time and effort in providing feedback on our manuscript. We have read and thoroughly considered your comments on including objective risk in the proposed work, and have made changes to the manuscript that we hope will address your concerns about the proposed work.

In line with your suggestion, we have included a measure of objective risk in the manuscript, and have chosen a measure that has been designed to assess people's risk of serious complications as a result of COVID-19 infection.

The measure is the Objective Risk Stratification tool (Jankowski et al., 2020), an existing risk assessment measure designed for healthcare workers. The items concern established risk factors for COVID-19, such as ethnicity, age, diabetes, pulmonary illness and cardiovascular disease. Additionally, the measure attributes different weights to illnesses, depending on their severity. For instance, having diabetes type 1 or 2 without complications has a score of 1, while diabetes type 1 or 2 with complications (i.e. acute or chronic health problems, such as eye, foot and kidney problems) results in a score of 2, as diabetes complications increase the risk of severe disadvantageous outcomes of COVID-19 infection. Final scores are the sum of weights across all items, with higher scores indicating higher risk of severe complications resulting from COVID-19 infection.

We believe that this measure is well-suited to the study, as the risk factors included in this measure are likely well-known to participants, and do not require specific details from medical records that we, and participants, cannot easily obtain. We believe this is the closest estimation we can make of one's risk of COVID-19 complications, within the scope of our project.

As a result, objective risk will replace subjective risk perception as a mediator in the planned analysis, and we will include subjective risk perception as part of the exploratory analysis section of our project instead. We have updated the introduction to include a section on objective risk, highlighting why we have decided to include this variable, and have updated the hypotheses, methods and analysis section of the manuscript.

We would like to thank you again for your time and effort spent improving the quality of our manuscript, and hope that this change is to your liking.

Reference

Jankowski, J., Davies, A., English, P., Friedman, E., McKeown, H., Rao, M., Sethi, S., & Strain, W. D. (2020). Risk Stratification tool for Healthcare workers during the CoViD-19 Pandemic; using published data on demographics, co-morbid disease and clinical domain in order to assign biological risk. *MedRxiv*, 2020.05.05.20091967. <https://doi.org/10.1101/2020.05.05.20091967>

Appendix C

July 16th, 2021

Dear reviewers,

Thank you again for providing feedback on our manuscript during the stage 1 revisions. We thoroughly considered your feedback and incorporated them into our study's design as much as possible, which we believe greatly benefitted our project. As a result, we welcome your suggestions and feedback on our Stage 2 submission, which includes the findings and conclusions of the project.

To summarize, the manuscript investigates adult age differences in coronavirus-related risk-taking, measured by the misalignment of government guidelines and adoption of preventative behaviours. To explain any potential age differences, we also included measures of objective risk of coronavirus-related illness and/or mortality, people's subjective perception of COVID-19 risk, their general attitude towards health risk, and people's numerical ability.

Our results show that older and younger adults differed in the adoption of preventative measures, with younger adults reportedly adopting preventative measures less frequently, thus taking more risk. This could be partly explained by numerical ability, which was higher in younger adults. Objective risk did not mediate the relationship between age group and COVID-19 risk-taking, though older adults were higher in objective risk for coronavirus-related illness or mortality, nor did people's general attitude towards health risk explain age differences in coronavirus-related risk-taking.

We also ran some exploratory analyses, which suggested some interesting findings. When taking overall, there were no age differences in subjective perception of COVID-19 risk, but younger adults did report perceiving more risk for both themselves and others when these were looked at separately. Both risk perception for others and oneself appeared to partly mediate the relationship between age group and COVID-19 risk-taking. In addition, when risk-taking items were split into governmental rules (i.e. mask wearing, social distancing) and recommendations (i.e. hand sanitizing), objective risk of coronavirus-related illness and/or death appeared to partially explain age differences in risk-taking when considering governmental rules (but not for general recommendations).

Taken overall, we believe our manuscript offers insight into behaviours during the coronavirus pandemic, specifically concerning age differences and any explanations for this, as well presenting some interesting findings that offer scope for future research in this area.

We would like to extend our thanks in advance for your time and effort in reviewing our work and hope you will find our Stage 2 submission to your liking.

With kind regards,

The authors of this manuscript

Appendix D

Response to reviewer

Dear Reviewer 2,

thank you for your time and effort spent reviewing our manuscript. It is much appreciated. We have taken your comments on board and implemented them in the manuscript whenever possible. You can find the summary of these changes in a response to each comment below.

With thanks,

The authors of this manuscript.

Comments

- 1. The results of this registered report do not confirm the initial hypotheses, namely that low numeracy would predict more “risk-taking” concerning Covid measures. There are two biases used as assumptions in this study that are highly problematic and persist in spite of the data : 1) the notion that “low-numeracy” people adhere less to Covid measures; 2) that the government, and specifically the UK government, provides scientifically accurate guidance on controlling the pandemic.**

Response:

1) We can see how this can be problematic. When deciding on our hypothesis concerning numeracy, we researched prior findings on numeracy, health behaviours and outcomes. Many studies showed a negative relationship between low numeracy and health behaviours/outcomes. For example, patients with low numeracy have been found to experience more difficulties in interpreting the risk of side effects (Gardner et al., 2011), more difficulties in understanding the information necessary to follow dietary requirements (Rothman et al., 2006), and make less accurate decisions about the risk of suffering a disease (Garcia-Retamero & Cokely, 2014; Petrova et al., 2016). A recent study on intentions to adopt preventive behaviours against COVID-19 and numeracy showed that higher approximate numeracy was related to higher intention (Sobkow et al., 2020). At the time of writing our hypotheses, we decided to base these on the findings of prior work on numeracy and health behaviours. However, our findings show a different pattern, as higher numeracy is related to lower adoption of preventive behaviours, and we have made changes so this is more thoroughly addressed in our discussion.

2) Concerning the scientific accuracy of the government guidelines, the government's approach to COVID-19 is informed by scientists; the Scientific Advisory Group for Emergencies (SAGE) advises the government on its coronavirus approach and receives external scientific advice. In addition to SAGE, the coronavirus guidelines given to English citizens at the time were identical, or similar, to those provided by the World Health Organisation (WHO, 2020).

In addition to the above, we discuss this topic more in-depth at point 5, as you provided further comments (with accompanying suggestions for improvement).

- 2. The exploratory analyses do not appear justified, methodologically, or informative. I advise the authors to, if they have a solid theoretical framework for one of these analyses, to better tighten their hypothesis and justification for it. Otherwise, should perform a follow-up study on separate cohort.**

Response: We agree with reviewer 2 that we have many exploratory analyses. We have gone through the exploratory analyses and removed all except the mediation analyses that concern risk perception. During the reviewing process of the stage 1 submission, the reviewers suggested replacing risk perception with objective risk. In order to still test the effect of risk perception, it was included in the exploratory section of the manuscript at the time. Our subsequent removal of (some of) the exploratory analyses have resulted in halving the analyses in the exploratory section.

- 3. Some of the data, especially those concerning numeracy and risk-taking, are very interesting; contra to the authors expectations; and deserve to be highlighted better than they are. On the other hand, the proliferation of qualitative risk measures muddy the picture of how risk affects behaviour. This aspect of the paper deserves a more careful and considered approach in the discussion.**

Response: Thank you for your comments and suggestions. We have made changes to the manuscript in line with your comments to discuss these aspects of the research in more detail, specifically to the abstract, results and discussion section of the manuscript.

- 4. Concerning the first point: Nowhere in the study – on risk no less – do the authors mention rationality or what constitutes low or high risk. Indeed, from the outset, it should have been clear that young people with high numeracy – that is people who can understand probabilities and percentages, and case fatality rates and so on - would adhere less to Covid measures because the risk of complications due to Covid remains low in the young. From my reading, the young act rationally – computing risk and guiding their actions accordingly. The finding (if read correctly – the specification of directionality in effects is unclear) that older people with low numeracy adhere less to Covid measures would seem to support the initial hypothesis concerning Covid and highlight a potential area to fix so to speak. But generally, the authors should question the very premise that less adherence to Covid measures actually constitutes risk-taking – or is it rational behavior?**

Response: Thank you for your suggestion. The study's results show that younger adults' lower adherence is partly explained by their higher numerical ability, which indicates that younger adults are able to calculate and understand their risk, and as such, adopt preventative behaviours less. However, they also reported perceiving higher risk for themselves (and for others), more so than older adults, and their risk perception also partially mediated their risk-taking. This finding is surprising, as their increased risk-taking does not appear to reflect less concern. Based on our study and the findings of others, we suspect that other factors related to this age group may explain this. For example, younger adults report higher levels of concern about their mental health and financial situation (i.e., reduced hours, losing income), which they may consider more detrimental than getting infected with COVID-19, or feel like they may not have a choice but to take the risk of infection. In the older age group, these factors have been reported to be much less of a concern (i.e., older adults were least worried about their mental health or finances compared to other age groups). As such, it seems likely that younger adults' lower adoption of preventative behaviours may be due to factors beyond numeracy and risk perception, which aren't currently investigated in this study but are considered in our discussion. In terms of whether the lack of adherence measures risk-taking; risk-taking is

generally considered as behaviour that increases the likelihood of harm to oneself or others. In this specific situation, not adhering to guidelines puts both the individual and others at risk. For example, not maintaining proper distancing means that a) the individual could be exposed to COVID-19, and b) they potentially expose another person to COVID-19 if the individual is infected. In addition to increasing the likelihood of illness or death (for oneself or others), not adhering to guidelines may also present another risk aside from illness and death, as it may lead to an increase in the duration of the pandemic and further disruption of one's day-to-day life. As such, we consider not adopting these preventative behaviours as risk-taking, as it increases the likelihood of COVID-19 infection (and any subsequent illness or death) for both the individual and others around them. Being younger may be associated with a lower likelihood of coronavirus-related illness and death, but the risk is not eliminated (i.e., though less often, young people can experience illness and death because of COVID-19, including long-term effects of COVID-19), and not adopting preventative behaviours also puts others at risk that younger people engage with, such as (older) family members. In addition, usage of the term risk-taking was approved during the stage 1 submission of this manuscript, and as such, alterations to the terminology would result in major changes to pre-approved sections of the manuscript.

- 5. Concerning the second point: The UK has one of the highest Covid mortality rates in OECD countries. In addition, its political body would appear to have gone through considerable volatility during the pandemic. As far as this reviewer understands, the UK health ministry is not led by a doctor or scientist. Thus, one is left to wonder why a citizen adhering to government mandates would be called risk-averse (in relation to health). In other words, by what measure is the UK government trustworthy? If that question is deemed inappropriate to this review, the authors should reconsider the equating of following UK government mandates with reasonableness/pro-sociality/rationality and/or risk aversion.**

Response: Thank you for this suggestion. You are correct concerning the UK Health Ministry being led by a non-scientist (Matt Hancock was the health secretary at the time, it is currently Sajid Javid). However, the government's approach to COVID-19 is informed by scientists; the Scientific Advisory Group for Emergencies (SAGE) advises the government on its coronavirus approach, and equally receives external scientific advice. The coronavirus guidelines given to English citizens at the time were identical, or similar, to those provided by the World Health Organisation (WHO, 2020). In addition, high mortality rates do not necessarily result from problematic guidelines (as these guidelines are similar to those applied elsewhere and are scientifically informed). Instead, the UK government's encouragement to follow these guidelines, the clarity of communication, and the timing of their implementation have been important to the uptake of preventative behaviours and preventing coronavirus-related illness and deaths and have been frequently criticized since the start of the pandemic.

For this study, we chose to include items that involved recommended preventative behaviours, some of which were also UK government guidelines, but not all. These were included because participants would be familiar with these preventative behaviours, instead of giving participants different behaviours than they were advised to adopt by their government, which could lead to confusion and affect participants' responses (for example, asking participants if they kept 2 meters apart could elicit a definite negative response despite their adherence to the 1-meter distance they were advised to maintain by their government).

Our main research question was how age groups differed in their adoption of preventative behaviours, and if this could be explained by numeracy, objective risk, risk preference, or risk perception. However, other research has found that trust in one's government may influence people's willingness to adopt behaviours. As such, we do see the merit in considering the trustworthiness of the government in terms of the adoption of preventative behaviours, but this is beyond the scope of our study. However, we have mentioned the findings of other studies on this in our discussion and think it would be worthwhile to research in the future.

- 6. While I welcome the addition of exploratory analyses, I feel that in this case they violate the spirit of registered reports. They read as fishing expeditions. All or most should be omitted from the study. The first recommendation concerns the analyses on government ordered vs recommended guidelines. I am not convinced this distinction is pertinent. It is also a tad provincial in that governments worldwide have drawn that distinction differently and therefore one cannot, based on these results, determine if differences are due to the quality of the guidelines (mandatory vs not) or to the guidelines themselves.**

Response: We agree with reviewer 2 that we have many exploratory analyses. We have gone through the exploratory analyses and removed all but the mediation analyses that concern risk perception, including the analyses on government guidelines and general recommendations.

- 7. The models concerning risk perception also appear to be very ad-hoc. Given that risk of Covid contraction; complication; and death exist as probabilities, why not ask respondents to state that probability (percentage) to compare to their objective risk? Respondents are left with ambiguous statements to rate. Then with regards to assessing risk to self and others, please refer to Covid-specific studies on the latter:
<https://doi.org/10.1002/ejsp.2737>
[10.31234/osf.io/qrbza](https://doi.org/10.31234/osf.io/qrbza)
<https://doi.org/10.1016/j.puhe.2020.06.032>**

Response: Thank you for this suggestion. We agree that this would be interesting to measure. At the time of our stage 1 submission, we decided to include this measure due to it having been peer-reviewed and adapted to specifically measure people's coronavirus risk perception. Due to the nature of registered reports, we are unable to alter the method at this time but will keep your suggestion in mind for any future work on the topic of risk perception and coronavirus risk.

- 8. Finally, given that the results don't confirm initial hypotheses, please: 1) emphasize this more clearly in the discussion; 2) streamline the writing. It is no longer clear exactly, given all the different results, what is interesting and why. There is an opportunity to do so in the discussion especially.**

Response: We have adapted our discussion in line with your suggestions and have expanded the section on numeracy and objective risk, in which directions are more thoroughly discussed, as well as possible reasons why these outcomes have been found.

- 9. Please review the text and streamline the language. There is considerable repetition (e.g. pp.45 and 46 on risk perception for self and others). While I do not agree with word limits, the manuscript is too long and could do with a heavy editing hand.**

Response: We have gone through the manuscript and have made sure to remove any repetitions as suggested (where possible).

- 10. Please check time indicators (are we still in a second wave, for instance?) and ensure that time-related terms remain coherent for a reader in the far-off future (e.g. make sure to include year, e.g. p.6, line 7).**

Response: Thank you for notifying us of this, we missed this at the time of submission. We have made changes to the document to make updates and remove any timestamps that might be confusing to the (future) reader, where possible.

- 11. Please make a table for all the risk measures listed (risk perception, objective risk, risk attitude, etc.) The field is large and nomenclature varies so it would be helpful to keep track of what's what.**

Response: Thank you for your suggestion. We have included a table with a list of variables and the materials used to measure them in the supplementary materials.

Suggestions

- 12. p.3 line 37, please provide a reference to the notion that adherence to, specifically, government-mandated directives enhances public health, especially given that you later divide public health measures into government-mandated versus recommended.**

Response: Unfortunately, we cannot add to this, as this section is part of the pre-approved Stage 1 submission.

- 13. p.7 line 57, please provide one or more recent references (the stability of risk preferences have been repeatedly challenged).**

Response: Unfortunately, we cannot change this, as it is part of the earlier approved Stage 1 manuscript. However, the stability of risk preference appears to be dependent on the type of measurement, as stated risk preference has been found to be stable over time, unlike revealed risk preference, as shown in Frey et al. (2017) and Frey et al. (2021).

- 14. p.8, line 55 Please provide directions in effects. What was the effect of numeracy in the Petrova et al. study, positive or negative?**

Response: In the study by Petrova et al. (2017), less numerate patients were about four times more likely to decide to delay critically needed medical treatment, increasing their risk for death and major disability. The estimated effect of numeracy was independent of many other cognitive, social, health, and demographic factors known to influence decision delay, such as age and symptom severity. Unfortunately,

we cannot add to this, as this section is part of the pre-approved Stage 1 submission.

15. p. 19. line 31, 32 is unclear.

Response: We have rewritten this section to make it clearer to the reader.

16. p. 19, final lines: why weren't these variables included in the final analysis? Or, alternatively, why were they included?

Response: Following our stage 1 registration protocol, we did not include these variables in the final analyses, we did collect this information for exploratory purposes only.

17. Figure 2: Can you add information to the caption or alternatively change or omit the figure? I see very little difference between the age groups.

Response: We agree that the figure shows little difference between the two groups, which aligns with the findings of the study. Where younger and older adults statistically differed, the difference was only small. We have noted this more clearly in the figure's caption and included a table with the means and standard deviations for both age groups in the supplementary materials (the reader is referred to this table through a footnote in the manuscript).

18. There remains a methodological issue concerning objective risk and age, It is clear, and confirmed, that these two variables are confounded (this paper lists an r of 0.72, which is on the stunning side in social science) so much so that they are almost the same.

Response: We followed the pre-registered protocol and used these variables as requested by a stage 1 reviewer, complemented with the mediation analyses, in which we replaced objective risk with subjective risk (as we originally proposed risk perception for our main analyses).

19. p.27 on numeracy and risk taking between age groups, please provide a test statistic on the numeracy item.

Response: As part of our planned analysis, we first tested the relationship between risk-taking and numeracy, followed by numeracy and age, and the mediation model. The description of numeracy, risk-taking and age is given to provide some background to the reader. The test statistic on age and numeracy is provided in the section further in the results, where the H2 hypotheses are tested.

20. p. 28 lines 49-53 Please specify direction of effects, especially given that you found an effect opposite to the one expected.

Response: Thank you for letting us know, we have added a sentence to clarify this.

21. Please include effect sizes where relevant.

Response: Thank you, we have included effect sizes for the two t-tests in our result section.

References

- Frey, R., Pedroni, A., Mata, R., Rieskamp, J., & Hertwig, R. (2017). Risk preference shares the psychometric structure of major psychological traits. *Science Advances*, 3(10), e1701381. <https://doi.org/10.1126/sciadv.1701381>
- Frey, R., Richter, D., Schupp, J., Hertwig, R., & Mata, R. (2021). Identifying robust correlates of risk preference: A systematic approach using specification curve analysis. *Journal of Personality and Social Psychology*, 120(2), 538–557. <https://doi.org/10.1037/pspp0000287>
- García-Retamero R., & Cokely E. (2014). The influence of skills, message frame, and visual aids on prevention of sexually transmitted diseases. *Journal of Behavioral Decision Making*, 27, 179–189. <https://doi.org/10.1002/bdm.1797>
- Gardner P. H., McMillan B., Raynor D. K., Woolf E., & Knapp P. (2011). The effect of numeracy on the comprehension of information about medicines in users of a patient information website. *Patient Education and Counseling*, 83(3), 398–403. <https://doi.org/10.1016/j.pec.2011.05.006>
- World Health Organisation. (2020, November 4). Critical preparedness, readiness and response actions for COVID-19. World Health Organisation. https://apps.who.int/iris/bitstream/handle/10665/336373/WHO-COVID-19-Community_Actions-2020.5-eng.pdf?sequence=1&isAllowed=y
- Petrova D., García-Retamero R., Catena A., & van der Pligt J. (2016). To screen or not to screen: What factors influence complex screening decisions? *Journal of Experimental Psychology: Applied*, 22(2), 247. <https://doi.org/10.1037/xap0000086>
- Rothman R. L., Housam R., Weiss H., Davis D., Gregory R., Gebretsadik T., ... & Elasy T. A. (2006). Patient understanding of food labels: The role of literacy and numeracy. *American Journal of Preventive Medicine*, 31, 391–398. <https://doi.org/10.1016/j.amepre.2006.07.025>